# Two Heads Are Better than One:
# Simulating Large Transformers with Small Ones

**Hantao Yu**
Department of Computer Science
Columbia University
New York, NY 10027
hantao.yu@columbia.edu

**Josh Alman**
Department of Computer Science
Columbia University
New York, NY 10027
josh@cs.columbia.edu

## Abstract

The quadratic complexity of self-attention prevents transformers from scaling effectively to long input sequences. On the other hand, modern GPUs and other specialized hardware accelerators are well-optimized for processing small input sequences in transformers during both training and inference. A natural question arises: can we take advantage of the efficiency of small transformers to deal with long input sequences?

In this paper, we show that transformers with long input sequences (large transformers) can be efficiently simulated by transformers that can only take short input sequences (small transformers). Specifically, we prove that any transformer with input length $N$ can be efficiently simulated by only $O((N/M)^2)$ transformers with input length $M \ll N$, and that this cannot be improved in the worst case. However, we then prove that in various natural scenarios including average-case inputs, sliding window masking and attention sinks, the optimal number $O(N/M)$ of small transformers suffice.

## 1 Introduction

The transformer architecture [VSP+17] has revolutionized modern machine learning, natural language processing and computer vision. It achieves state-of-the-art performance on various tasks such as language reasoning [Dee21, Ope20], image recognition [KDW+21, CMS+20] and many others. At the core of the transformer architecture is the attention mechanism, which captures correlations between all pairs of tokens. However, this is also a major bottleneck for transformers, as the quadratic complexity (in both time and memory) of the attention mechanism prohibits effective scaling of transformers as the sequence grows in length. Moreover, it has been theoretically proved that the quadratic complexity cannot be avoided (under popular complexity-theoretic assumptions) [AS24]. To address this fundamental issue, there has been a fruitful literature on the design of "subquadratic alternatives" to transformers, where researchers come up with mechanisms that replace the attention mechanism and take subquadratic time (usually close to linear time) [KKL20, CLD+21, DFE+22, KMZ24, BPC20, GD23]. However, they usually have worse performance than standard transformers, especially on downstream tasks and translation [VPSP23, JBKM24, AY25].

In the meantime, modern GPUs are increasingly optimized for handling short-to-moderate transformer contexts [WXQ+21, DFE+22]. Some companies are even producing specialized hardware for efficient transformer inference that have superior performance on inputs of length between 128 to 2048 [Kim24, Etc24]. This approach motivates the following questions:

*Can we use small transformers to perform tasks more efficiently than large transformers? Are (multiple) small transformers inherently capable of dealing with long contexts?*

39th Conference on Neural Information Processing Systems (NeurIPS 2025).

In this paper, we give positive answers to these questions through the lens of representational strength, which studies whether one can select parameters for transformers so that they can perform certain tasks of interest. The representational strength of transformers has been studied broadly in recent years [BHBK24, LAG$^+$23, SHT24, MS23], and it is believed that it is one of the core reasons why transformers outperform previous architectures such as RNN and LSTM [WDL25, AET$^+$24, SHT23].

Our problem can be stated as follows. Suppose that we have all the parameters of a large transformer $\mathcal{T}$ with input length $N$, as well as an input $X$ that we would like to evaluate $\mathcal{T}$ on. However, to evaluate $\mathcal{T}(X)$, we are not allowed to perform any particularly complicated computations; we are restricted to simple operations, and to making use of a small transformer $\mathcal{O}$ (as an oracle) that can only take input sequences of length $M \ll N$. We can input into $\mathcal{O}$ any sequence and parameters that we can easily compute, and obtain its output. Our goal is to minimize the number of calls to $\mathcal{O}$ that we need to obtain $\mathcal{T}(X)$ for arbitrary input $X$ of length $N$.

Our main results show that roughly $O((N/M)^2)$ oracle calls suffice, which we show is optimal. In addition, our algorithm requires minimal processing outside of the oracle calls, and it has properties needed for efficient training and inference, including that the gradients of its parameters are easily computed, and that its oracle calls can be computed in parallel in only $O(L)$ rounds of adaptivity, where $L$ is the number of layers in the large transformer $\mathcal{T}$.

In addition, we show that in many scenarios arising in practice, such as when certain masking schemes are used, or when the data is not "worst-case" and satisfies some boundedness guarantees, the information-theoretically optimal $O(N/M)$ oracle calls suffice.

Our results provide a new way to deal with long input sequences for transformers, as we prove that any computation performed by large transformers can be decomposed into computations that only use smaller transformers. If the oracles are implemented using a quadratic number of floating-point operations, then our algorithm also still requires a quadratic amount of floating-point operations. However, if modern GPUs enable faster transformer inference with respect to the "wall-clock" time when the input sequence is short-to-moderate, then our algorithms allow faster wall-clock time inference. For example, if the oracle can compute the output using $O(M)$ wall-clock time compared to the standard $O(M^2)$ time, then the total wall-clock running time of our algorithms will be $O(N^2/M)$.

Our approach is fundamentally different from designing "subquadratic alternatives" to transformers [KKL20, CLD$^+$21, BPC20, KMZ24, GD23]. In particular, our algorithm preserves the representational strength of transformers (or even improves it), whereas it has been shown that all the subquadratic alternatives to transformers will lose representational strength as they cannot capture all the pairwise relationship even approximately [AY25].

Now we define our model of computation and state our main contributions in more detail.

## 1.1 Computational Model for Simulating Large Transformers

We now describe our model of computation in more detail. We are careful to allow only very simple operations beyond oracle calls, to ensure that the vast majority of computation can be performed by efficient hardware for evaluating small transformers, and that the number of oracle calls accurately measures the complexity of the problem.

We are given a large transformer $\mathcal{T}$ with input length $N$, $L$ layers, $H$ attention heads in each layer, and embedding dimension $d$ (all the parameters, including the query, key, value matrices in each of its attention head and multilayer perceptron functions). Throughout this paper, we assume that $L, H \ll N, d = O(\log N), \Omega(\log N) \le M < o(N)$, and one can typically imagine $M \approx \sqrt{N}$. Our goal is to design an algorithm that (approximately) output $\mathcal{T}(X) \in \mathbb{R}^{N \times d}$ for arbitrary input $X$ (length at most $N$).

We have a limited set of operations we can perform as part of the algorithm. We critically have access to a small transformer (oracle) $\mathcal{O}$ that can take as input a sequence of length at most $M \ll N$, as well as the parameters for a transformer which has $L'$ layers and $H'$ attention heads in each layer, and outputs the transformer evaluated on that sequence. Our algorithm is allowed to:

1. Feed the oracle $\mathcal{O}$ with input sequences and parameters which are currently in memory to obtain its output;

2. Processing: Edit existing vectors or matrices in memory by padding at most $O(d^2)$ fixed numbers (constants independent of the input) to them, or arranging (concatenating) matrices in memory.

We also assume that all the numbers in input matrices, parameters, and algorithms have $O(\log N)$-bit representations. We say that such an algorithm *simulates* $\mathcal{T}$ if it always outputs a $Y \in \mathbb{R}^{N \times d}$ such that

$$\|Y[i,:] - \mathcal{T}(X)[i,:]\|_2 \le \varepsilon \|\mathcal{T}(X)[i,:]\|_2$$

for all $i \in [N]$, and for very small error $\varepsilon = \Theta(\frac{1}{2^N})$. We want to design algorithms that simulate $\mathcal{T}$ with as fewer oracle calls as possible. (Such an $\varepsilon$ is essentially unavoidable in limited precision architectures, but we will see it will be very helpful in some algorithms below. We also emphasize that our main result, Theorem 1.1, is an exact computation in the unlimited precision scenario with $\varepsilon = 0$.)

Notice that any such algorithm can be viewed as a composition of oracles and the padding function. Since we only allow for very simple processing, it is straightforward to compute the gradients of the padding functions, so training the large model could be done via computing the gradients of the small transformer oracles.

## 1.2 Main Results

**Quadratic small transformers are sufficient and necessary for worst-case inputs.** As summarized below, our main result shows that any computation performed on a large transformer can be decomposed into multiple instances of computation performed on smaller transformers with the same computational complexity or floating-point operations. Since the oracle can only tell us the final output instead of intermediate embeddings, it might be somewhat surprising that we are able to utilize all the layers in small transformers.

**Theorem 1.1** (Theorem 3.4, Theorem 3.5). *For any transformer $\mathcal{T}$ with $L$ layers, $H$ attention heads in each layer, input length $N$, embedding dimension $d$, there exists an algorithm that simulates $\mathcal{T}$ with $O((\frac{N}{M})^2 \cdot \frac{HL}{H'L'})$ calls to a transformer oracle with $L'$ layers, $H'$ attention heads in each layer, input length $M$, embedding dimension $O(\frac{dH'L'}{H})$. The result still holds when we add causal masking to both large and small transformers.*

Notice that these simulations are *tight*, and roughly $(N/M)^2$ oracle calls are necessary in the worst-case due to computational complexity constraints. To see this, note that when $L = H = L' = H' = 1$, a straightforward algorithm can compute the responses of $T$ oracle calls in time only $\tilde{O}(TM^2)$. Thus, since it is known that even approximation of a large attention requires time $\Omega(N^{2-o(1)})$ (under standard complexity-theoretic assumptions) [AS24], we must have $T \ge ((N/M)^2)^{1-o(1)}$.

One might be concerned with the fact that $O((N/M)^2)$ small transformers have many more parameters than one large transformer, since each transformer has $\Theta(d^2)$ parameters, independent of the sequence length. However, this is not a problem because in our construction, we reuse the parameters such that the total number of parameters does not depend on $N$. In fact, all the query, key and value matrices that we feed into the oracle share most entries with the query, key, value matrices in the large transformer that are given. We ultimately only have a small, constant factor blowup on the number of parameters.

**Linear small transformers are weaker but sufficient with average-case inputs.** Even though we cannot use $O(N/M)$ oracle calls to simulate a large transformer in the worst case, we show that it is possible when we have reasonable additional assumptions on the queries, keys and values.

**Theorem 1.2** (Informal version of Theorem 4.1). *Let $\mathcal{T}$ be a transformer with $L$ layers, $H$ attention heads in each layer, input length $N$ and embedding dimension $d$. Suppose that the queries, keys and values in the attention heads are all somewhat bounded in how much they may differ from each other. Then, there exists an algorithm using $O(\frac{N}{M} \cdot \frac{HL}{H'L'})$ oracle calls to simulate $\mathcal{T}$.*

Models such as Hierarchical Transformers [PZV$^+$19, LL19, CDF$^+$22] split the input sequence into chunks of size $M$ and send each chunk into a transformer before aggregating the outputs. We give the first provable guarantees for this approach, showing that $O(N/M)$ small transformers have approximately equal expressivity as a large transformer when the input data satisfies our assumptions.

This provides a possible explanation of the success of Hierarchical Transformers and relevant ideas from an expressivity viewpoint.

On the other hand, we supplement Theorem 1.2 with its converse, which shows that a linear number of small transformers are at most as expressive as one large transformer for worst-case inputs (we only prove the statement for single-head transformers to illustrate the message). As a result, when the inputs follow assumptions in Theorem 1.2, a linear number of small transformers are *equivalent* to a larger one in expressive power.

**Theorem 1.3** (Theorem 4.2). *Given $N/M$ instances of single layer, single head transformers with input length $M$ and embedding dimension $d$, there exists an algorithm that simulates them with one call of a single layer, single head transformer with input length $O(N)$ and embedding dimension $O(d)$, along with $O(N/M)$ many matrix multiplications of size $M \times d \times d$.*

We briefly comment on the small matrix multiplications in Theorem 1.3. Note that they could be computed in the straightforward way in nearly linear time $O(Md^2)$ (since $d = O(\log N)$) and thus do not substantially contribute to the total running time. This implies, in particular, that they could not simulate the transformer oracles on their own, and are only "assisting" the large transformer oracle. Their presence seems unavoidable because of the $\Theta(N/M)$ weight matrices of the oracles which must be simulated by a single large transformer, which only has a constant number of weight matrices. Moreover, we emphasize that our other constructions are even simpler, and do not need such small matrix computations outside of the oracle calls.

**Efficient simulation of transformers with sliding window and StreamingLLMs.** Sliding window and StreamingLLM [XTC$^+$24] are popular ways to make transformer inference more memory efficient. Both sliding window and StreamingLLM are based on the observation that certain attention scores are often higher than others. Sliding window is based on the intrinsic structure of languages, where each token is typically more correlated to the previous few tokens. Therefore, for each query we only take into account the contributions of the keys that are positionally close to it. The StreamingLLM framework is motivated by the observation that autoregressive LLMs have a surprisingly large amount of attention score concentrated to the initial tokens, and thus each query only takes into account keys that are positionally close to it, as well as the first few (usually $3 \sim 5$) keys, which are called "attention sinks".

We show that in both cases we can use a linear number of small transformer oracle calls to simulate them, even in the worst case. As summarized below, our result indicates that oracles can capture efficient attention based on sliding windows and attention sinks efficiently.

**Theorem 1.4** (Theorem 5.1). *For any transformer $\mathcal{T}$ with $L$ layers, $H$ attention heads in each layer, input length $N$, embedding dimension $d$, constant-size sliding window, there exists an algorithm that simulates $\mathcal{T}$ with $O(\frac{N}{M} \cdot \frac{HL}{H'L'})$ calls to a transformer oracle with $L'$ layers, $H'$ attention heads in each layer, input length $M$ and embedding dimension $O(\frac{dH'L'}{H})$ with causal masking. This result still holds if we have constant-size attention sinks.*

## 1.3 Related Work

**Representational strength and limitations of transformers.** The representational strength of transformers has been intensively studied in recent years from a variety of perspectives. To list a few, [MSS22, MS23, SMW$^+$24] study the class of problems that transformers can solve from a circuit complexity viewpoint; [BAG20, LAG$^+$23, Hah20] aim to understand whether transformers can recognize formal languages; [SHT23] focus on reasoning tasks and show that transformers are inherently capable of solving sparse averaging; [SHT24] gives important connections between transformers and the massively parallel computation model; [HSK$^+$25, HWL$^+$24] uses computational complexity to characterize the computational limits of diffusion transformers and low-rank adaptation for transformers; [LCW23] studies attention's capability of approximating sparse matrices; [YBR$^+$20, YCB$^+$20] shows that transformers and many subquadratic variants are universal approximators for sequence-to-sequence functions;

**Fast attention mechanisms.** There has been a fruitful literature of dealing with long input sequence by designing "subquadratic alternatives" to transformers, which are variants on the attention mechanism which can be performed in subquadratic time. For example, researchers have studied various sparse attention mechanisms that only consider the query-key pairs that have high correlation,

including Reformer [KKL20], Longformer [BPC20], and Hyperattention [HJK+24]. Additionally, there has been work on kernel/low-rank attention that approximates attention mechanism using kernels such as Performer [CLD+21] and Polysketchformer [KMZ24], and there has been a growing interest in state space models such as Mamba [GD23]. See [TDBM22] for a comprehensive survey on efficient attention mechanisms. However, [AY25] proves that none of these subquadratic models can capture all pairwise correlations even approximately as the sequence length grows.

## 2 Preliminaries

### 2.1 Transformers

We first define the standard attention mechanism in Transformer.

**Definition 2.1** (Attention Mechanism). *Given input* $X \in \mathbb{R}^{N \times d}$, *query, key, value matrices* $W^Q, W^K, W^V \in \mathbb{R}^{d \times m}$, *the* attention mechanism *computes*

$$\text{Attn}(X) = \text{softmax}((XW^Q)(XW^K)^\top)(XW^V) \in \mathbb{R}^{N \times m}.$$

Here we say $N$ is the *context length*, and $d$ is the embedding dimension. We will also call each attention mechanism an *attention head* in the transformer architecture. For notational convenience, we let

$$\{q_1, \ldots, q_N\} \in \mathbb{R}^m, \{k_1, \ldots, k_N\} \in \mathbb{R}^m, \{v_1, \ldots, v_N\} \in \mathbb{R}^m$$

be the rows of $XW^Q, XW^K, XW^V$ respectively. We will call them the queries, keys and values. As a result, the attention mechanism is computing

$$\frac{1}{\sum_{j=1}^N \exp(\langle q_i, k_j \rangle)} \sum_{j=1}^N \exp(\langle q_i, k_j \rangle) \cdot v_j$$

for each query $q_i$.

Another important component in transformers is the *multilayer perceptron* (MLP). An MLP is a feed-forward, fully-connected neural network consisting of one or more hidden layers using ReLU activation. The universal approximation theorem states that any continuous function with a finite support can be approximated by a neural network with one hidden layer [HSW89]. In light of this, in many relevant works [SHT23, SHT24], MLPs are modeled as arbitrary functions on compact domains.

In this paper, our goal is to use small transformers to simulate large transformers, and we would like to ensure that the MLPs in the small transformers are as simple as possible. We will therefore assume that MLPs in small transformers compute functions $\phi : \mathbb{R}^d \to \mathbb{R}^{O(d)}$ such that:

1. They are at least as strong as the MLPs in the large transformers, i.e. they can do whatever computation that MLPs in the large transformers can do, and

2. They can do basic arithmetic operations on the input vector $x \in \mathbb{R}^d$ or pad fixed numbers to it (both are simple continuous functions) as long as they take $O(d^2)$ time.

When a MLP $\phi$ is applied on a matrix, it will be applied row-wise to output another matrix. In other words, it is applied on each token given a sequence of tokens.

An *attention layer* $f$ with $H$ attention heads consists of $H$ attention mechanisms with query embedding $W_h^Q, W_h^K, W_h^V \in \mathbb{R}^{m \times m}$ for the $h$-th attention such that $m = \frac{d}{H}$. The input $X$ is partitioned into $H$ matrices $X[:, D_1], \ldots, X[:, D_H] \in \mathbb{R}^{N \times m}$ column-wise, where $D_i = \{\frac{(i-1)d}{H} + 1, \ldots, \frac{id}{H}\}$ for all $i$, such that the $h$-th attention head computes

$$\text{softmax}((X[:, D_h]W_h^Q)(X[:, D_h]W_h^V)^\top)(X[:, D_h]W_h^V) \in \mathbb{R}^{N \times m}.$$

All attention outputs are concatenated column-wise and fed through a *layer MLP* $\psi$ such that the output of attention layer $f$ is

$$f(X) := \psi\Big( \big[ \text{softmax}((X[:, D_h]W_h^Q)(X[:, D_h]W_h^V)^\top)(X[:, D_h]W_h^V) \big]_{h=1}^H \Big) \in \mathbb{R}^{N \times d}.$$

**Definition 2.2** (Transformer). *A transformer $\mathcal{T}$ with $L$ layers and $H$ attention heads in each layer consists of an input MLP $\phi : \mathbb{R}^{d'} \to \mathbb{R}^d$ applied token-wise on the input $X \in \mathbb{R}^{N \times d'}$, $L$ attention layers $f_1, \ldots, f_L : \mathbb{R}^{N \times d} \to \mathbb{R}^{N \times d}$ which contain $L$ layer MLPs $\psi_1, \ldots, \psi_L$ applied token-wise at the end of each attention layer. For each $2 \leq \ell \leq L$,*

$$X^{(1)} = \psi_1(f_1(\phi(X))), X^{(\ell)} = \psi_\ell(f_\ell(X^{(\ell-1)})).$$

*Finally, the transformer $\mathcal{T}$ outputs $\mathcal{T}(X) = X^{(L)}$.*

We will simplify the notion of positional encoding into input MLP $\phi$ and assume that the input MLP has positional information of the tokens. In other words, if $x_i = X[i, :]$ is the $i$-th input token, then $\phi(x_i)$ is also a function of $i$.

Transformer is powerful as computational model, and we refer the readers to Appendix A for more operations that transformers can do that will be useful in our proofs.

**Transformer Oracle.** A *transformer oracle $\mathcal{O}$* is a small transformer that can only take inputs of length at most $M$ (its embedding dimension, number of layers, number of heads in each layer, causal masking etc will be specified in result statements). In this paper we are mostly concerned with simulating transformers with large input length $N$ using transformer oracles with input length $M$ such that $M \ll N$ (recall that we assume $\Omega(\log N) \leq M < o(N)$).

## 2.2 Causal masking, sliding window and StreamingLLM

We first define the most commonly used causal masking in attention heads.

**Definition 2.3** (Causal Masking Attention). *Given input $X \in \mathbb{R}^{N \times d}$, query, key, value matrices $W^Q, W^K, W^V \in \mathbb{R}^{d \times m}$, the attention mechanism with causal masking computes*

$$\text{Attn}(X) = \text{softmax}\Big(\text{mask}((XW^Q)(XW^K)^\top)\Big)(XW^V) \in \mathbb{R}^{N \times m},$$

*where the mask function sets all upper triangular entries (not including diagonal entries) to $-\infty$.*

Another commonly used masking scheme for efficient transformer inference/training is sliding window, where we only keep the keys whose indices are close to the query index.

**Definition 2.4** (Sliding Window Attention). *Given input $X \in \mathbb{R}^{N \times d}$, query, key, value matrices $W^Q, W^K, W^V \in \mathbb{R}^{d \times m}$ and window size $r \geq 1$, the attention mechanism with sliding window of size $r$ computes*

$$\text{Attn}(X) = \text{softmax}\Big(\text{window}((XW^Q)(XW^K)^\top)\Big)(XW^V) \in \mathbb{R}^{N \times m},$$

*where the window function sets all entries in $\{(i, j) : j > i \text{ or } j \leq i - r\}$ to $-\infty$.*

In other words, for each query $q_i$ we only look at $k_j$ such that $i - r + 1 \leq j \leq i$.

Finally, StreamingLLM [XTC+24] is a framework designed for efficient training with a finite length window. Upon having a fixed-size sliding window, each query also attends to the first $s$ keys (called "attention sinks"), where $s$ is usually a small positive constant (around $3 \sim 5$).

**Definition 2.5** (Attention Sink). *Given input $X \in \mathbb{R}^{N \times d}$, query, key, value matrices $W^Q, W^K, W^V \in \mathbb{R}^{d \times m}$ and window size $r \geq 1$, sink size $s \geq 1$, the attention mechanism with attention sink computes*

$$\text{Attn}(X) = \text{softmax}\Big(\text{sink}((XW^Q)(XW^K)^\top)\Big)(XW^V) \in \mathbb{R}^{N \times m},$$

*where the sink function sets all entries in $\{(i, j) : j > i \text{ or } s < j \leq i - r\}$ to $-\infty$.*

Transformers with causal masking attention, sliding window, and StreamingLLMs are defined exactly the same as transformers except that we replace standard attention mechanisms by attention with causal masking, attention with sliding window and attention with sinks.

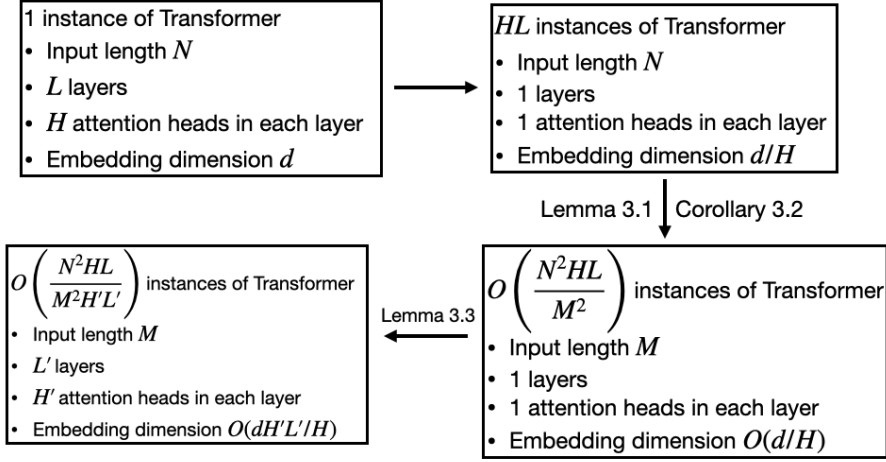

Figure 1: Proof Roadmap

## 2.3 Notation

Throughout the paper, we denote $X \in \mathbb{R}^{N \times d}$ as the input to the large transformer, where $N$ is the input length and $d$ is the embedding dimension. For a $N \times d$ matrix $X$, we use $X[i,:]$ to denote its $i$-th row, $X[:,j]$ to denote its $j$-th column, and $X[i,j]$ to denote its $(i,j)$-th entry. Given sets $S \subseteq [N], D \subseteq [d]$, we use $X[S,:]$ to denote the submatrix consisting of the rows in $S$, $X[:,D]$ to denote the submatrix consisting of the columns in $D$, and $X[S,D]$ to denote the submatrix consisting of the entries in $S \times D$ of $X$.

We use $W^Q, W^K, W^V$ to denote the query, key and value matrices for attention heads, and we use $q_i, k_i, v_i$ to denote the $i$-th row of $XW^Q, XW^K, XW^V$ respectively. We also let $S_t = \{(t-1)M + 1, tM\}$ for all $1 \le t \le N/M$.

We use $\mathbf{1}_{a \times b}$ to denote the $a \times b$ matrix whose entries are all 1, and $\mathbf{0}_{a \times b}$ to denote the $a \times b$ matrix whose entries are all 0.

We now turn to proving our results. We give proof sketches and main ideas here; full proofs are deferred to the appendix.

## 3 Quadratic calls suffice for simulation

In this section, we prove that $O((\frac{N}{M})^2 \cdot \frac{HL}{H'L'})$ small transformers with $L'$ layers and $H'$ attention heads in each layer suffice to simulate a large transformer with $L$ layers and $H$ attention heads in each layer (Theorem 3.4). Our proof roadmap is illustrated in Figure 1, where the arrows $A \to B$ indicate that $A$ can be simulated by $B$. Complete proofs of all the statements can be found in Appendix B. We first show that this is the case when they both only have a single attention head, i.e $H' = H = L = L' = 1$.

**Lemma 3.1.** *For any single layer, single head transformer $\mathcal{T}$ with input length $N$, embedding dimension $d$, there exists an algorithm that simulates $\mathcal{T}$ with $O(\frac{N^2}{M^2})$ calls to a transformer oracle with input length $M$ and embedding dimension $O(d)$.*

*Proof Sketch.* Our high-level idea is to partition the $N \times N$ attention matrix of $\mathcal{T}$ into $\frac{N^2}{M^2}$ blocks of size $M \times M$, and then use a constant number of oracles to separately handle each block. In particular, each block corresponds to two sub-intervals of length $M$ out of the input sequence of length $N$ (one interval for the rows, or queries, and one interval for the columns, or keys), so we can aim to have an oracle for sequence length $O(M)$ compute the contribution of each block. To be more precise, as in Definition 2.1 above, let

$$\{q_1, \ldots, q_N\} \in \mathbb{R}^m, \{k_1, \ldots, k_N\} \in \mathbb{R}^m, \{v_1, \ldots, v_N\} \in \mathbb{R}^m$$

be the rows of $XW^Q, XW^K, XW^V$ respectively. Define

$$a_{i,j} = \exp(\langle q_i, k_j \rangle), \text{ and } b_{i,j} = \exp(\langle q_i, k_j \rangle) \cdot v_j.$$

The goal of the attention mechanism is to compute, for all $i$,

$$\frac{1}{\sum_{j=1}^N \exp(\langle q_i, k_j \rangle)} \cdot \sum_{j=1}^N \exp(\langle q_i, k_j \rangle) \cdot v_j = \frac{\sum_{j=1}^N b_{i,j}}{\sum_{j=1}^N a_{i,j}} = \frac{\sum_{t=1}^{N/M} \sum_{j \in S_t} b_{i,j}}{\sum_{t=1}^{N/M} \sum_{j \in S_t} a_{i,j}}.$$

The main technical difficulty is that one oracle call is not able to give us information on the sum of $a_{i,j}$ (or $b_{i,j}$) over all $j \in [N]$. However, we show that one oracle call allows us to compute $\sum_{j \in S_t} a_{i,j}$, where $S_t = \{(t-1)M + 1, \ldots, tM\}$ such that $N/M$ oracle calls suffice to give us $\sum_{j=1}^N a_{i,j}$. We do this by adding in one synthetic token to the sequence so that its contribution is fixed, i.e. its inner product with all the keys will be the same and known. In addition, we assign its corresponding value token to be $0$ and other value tokens to be $1$ such that the output does not contain the synthetic token's value, while the normalizing term still counts its contribution. As a result, the output of the oracle will give us

$$\frac{\sum_{j \in S_t} a_{i,j}}{\sum_{j \in S_t} a_{i,j} + a},$$

where $a$ is the attention value (that we set and thus know in advance) for the normalizing term. This information allows us to compute $\sum_{j \in S_t} a_{i,j}$ as we can solve a linear equation using MLP. Secondly, we will directly feed the oracle with $X[S_t, :], W^Q, W^K, W^V$ to obtain

$$\frac{\sum_{j \in S_t} b_{i,j}}{\sum_{j \in S_t} a_{i,j}},$$

and since we already know $\sum_{j \in S_t} a_{i,j}$, we can compute $\sum_{j \in S_t} b_{i,j}$, which will furthermore give us $\sum_{j=1}^N b_{i,j}$ by summing them up. $\qquad\square$

We now move on to generalizing Lemma 3.1 to general $H, L, H', L'$, i.e., when the transformer $\mathcal{T}$ and the oracles can have multiple heads and layers. A first attempt to do this might use different layers of $\mathcal{O}$ to simulate different layers of $\mathcal{T}$, but this appears difficult to implement, since Lemma 3.1 requires some processing between layers of attentions that is not available to us when the different layers are connected only through MLPs within an oracle. Indeed, since the output of each attention head needs processing before it can be used in another attention head, it is unclear how to take advantage of more than one layer of each oracle in this way. We instead take a different approach: we use all the attention heads in all the layers of the oracle completely independently from each other to simultaneously simulate $\Theta(H'L')$ different attention heads, and we use these all together to simulate one layer at a time of $\mathcal{T}$.

First, it is not hard to generalize Lemma 3.1 to general $H, L$ (but still $H' = L' = 1$) by separately simulating each attention head in $\mathcal{T}$ regardless of which layer it is in:

**Corollary 3.2.** *For any transformer $\mathcal{T}$ with $L$ layers, $H$ attention heads in each layer, input length $N$, embedding dimension $d$, there exists an algorithm that simulates $\mathcal{T}$ with $O((\frac{N}{M})^2 \cdot HL)$ calls to a single head, single layer transformer oracle with input length $M$ and embedding dimension $O(\frac{d}{H})$.*

Next we show that a transformer with $L'$ layers, $H'$ attention heads in each layer, input length $M$, and embedding dimension $O(\frac{dH'L'}{H})$ can be used to simulate $H'L'$ instances of single head, single layer transformers with input length $M$ and embedding dimension $\frac{d}{H}$. In other words, we are able to independently use each attention head in the transformer, regardless of which of the $L'$ layers it appears in:

**Lemma 3.3.** *One transformer with $L'$ layers, $H'$ attention heads in each layer, input length $M$ and embedding dimension $O(\frac{dH'L'}{H})$ can be used to simulate $H'L'$ independent instances of single layer, single head transformers with input length $M$ and embedding dimension $\frac{d}{H}$.*

*Proof Sketch.* Consider first when $L' = 1$. A transformer with $H'$ attention heads naturally partitions (by definition) the embedding dimension $O(\frac{dH'}{H})$ into $H'$ parts of size $O(\frac{d}{H})$, and each head

separately computes an attention mechanism on one of those parts. The result follows almost directly, with some care to details about MLPs and aggregation.

More care is needed when $L' > 1$. We partition the $\Theta(\frac{dH'L'}{H})$ coordinates of the embedding dimension into $L'$ parts of size $\Theta(\frac{dH'}{H})$ each, and the key idea is that each layer will operate on one of those parts while leaving the rest unchanged. Indeed, weights for the query and keys can be selected so that only the relevant part of the coordinates will impact the attention matrices at each layer, then weights for the values can be selected so that the other parts are passed through the layer without being changed. $\qquad\square$

Finally, we combine Lemma 3.1, Corollary 3.2 and Lemma 3.3 to obtain our main result.

**Theorem 3.4.** *For any transformer $\mathcal{T}$ with $L$ layers, $H$ attention heads in each layer, input length $N$, embedding dimension $d$, there exists an algorithm that simulates $\mathcal{T}$ with $O((\frac{N}{M})^2 \cdot \frac{HL}{H'L'})$ calls to a transformer oracle with $L'$ layers, $H'$ attention heads in each layer, input length $M$, embedding dimension $O(\frac{dH'L'}{H})$.*

We additionally show that these results still hold when both the large and small transformers have causal masking. The proofs are similar to the proof of Theorem 3.4, and are deferred to Appendix B.

**Theorem 3.5.** *For any transformer $\mathcal{T}$ with $L$ layers, $H$ attention heads in each layer, input length $N$, embedding dimension $d$ and causal masking, there exists an algorithm that simulates $\mathcal{T}$ with $O((\frac{N}{M})^2 \cdot \frac{HL}{H'L'})$ calls to a transformer oracle with $L'$ layers, $H'$ attention heads in each layer, input length $M$, embedding dimension $O(\frac{dH'L'}{H})$ and causal masking.*

## 4 Efficient simulation with average-case input assumptions

### 4.1 Linear calls suffice for average-case inputs

In this section we prove that if the queries, keys and values in the attention heads are somewhat bounded in how much they may differ from each other, then $O(\frac{N}{M})$ small transformers suffice to approximate a large transformer. We provide a proof sketch below, and the complete proof can be found in Appendix C.1.

**Theorem 4.1.** *Let $\mathcal{T}$ be a transformer with $L$ layers, $H$ attention heads in each layer, input length $N$ and embedding dimension $d$. Suppose there exist absolute constants $C, D > 0$ such that*

$$\frac{1}{C} \le a_{i,j} \le C, \text{ and } DN \cdot \max_j \|b_{i,j}\|_2 \le \Big\| \sum_{j=1}^{N} b_{i,j} \Big\|_2$$

*where*

$$a_{i,j} = \exp(\langle q_i, k_j \rangle), b_{i,j} = \exp(\langle q_i, k_j \rangle) \cdot v_j$$

*for any query $q_i, k_j, v_j$ in any attention head. There exists an algorithm using $O(\frac{N}{M} \cdot \frac{HL}{H'L'})$ oracle calls to a small transformer with $L'$ layers, $H'$ attention heads in each layer, embedding dimension $O(\frac{dH'L'}{H})$ to obtain an $(1 + \varepsilon)$ approximation of $\mathcal{T}$ with probability at least $0.9$ for any fixed constant $\varepsilon > 0$.*

*Proof Sketch.* The high-level idea is to partition the queries into $N/M$ parts of size $M$ each, and then permute the keys (but not the queries) using a random permutation $\tau$. We then aim to approximate the desired quantities $\sum_{j=1}^{N} \exp(\langle q_i, k_j \rangle)$ and $\sum_{j=1}^{N} \exp(\langle q_i, k_j \rangle) \cdot v_j$ using rescalings of $\sum_{j \in S_t} \exp(\langle q_i, k_{\tau(j)} \rangle)$ and $\sum_{j \in S_t} \exp(\langle q_i, k_{\tau(j)} \rangle) \cdot v_{\tau(j)}$ (where $S_t$ is the part of size $M$ that contains query $q_i$). This can be seen as estimating the desired sums by sampling only $M$ of the $N$ summands at random. At the same time, by blocking the queries and keys like this, the samples can be computed using oracle calls similar to Lemma 3.1 above. $\qquad\square$

### 4.2 Linear small transformers are weaker than large transformers

We also show that $N/M$ small transformers can be simulated by a large transformer along with an oracle for performing very small matrix multiplications ($M \times d$ with $d \times d$). We only prove the statement for single head transformers to illustrate the need for linear amount of oracle calls.

**Theorem 4.2.** *Given $N/M$ instances of single layer, single head transformers with input length $M$ and embedding dimension $d$, there exists an algorithm that simulates them with one call of a single layer, single head transformer with input length $O(N)$ and embedding dimension $O(d)$, along with $O(N/M)$ many matrix multiplications of size $M \times d \times d$.*

The key idea behind Theorem 4.2 is to concatenate the tokens from all $N/M$ input sequences into a single long sequence of length $N$, but then slightly increase the embedding dimension in a way which makes tokens from different short sequences highly uncorrelated with each other. Thus, the large attention will not give much weight to pairs of tokens from different short sequences. The complete proof is delayed to Appendix C.2.

## 5 Simulation of transformers with sliding window and StreamingLLMs

In our last section, we show that small transformers work well when we add sliding window masking to attention heads, and when in the StreamingLLM framework. Our constructions are similar to above, but with additional techniques to take advantage of the masking structures, and can be found in the Appendix D.

When we consider sliding window attention, we only need to deal with keys that are close to each query. It is intuitive to see that in such scenario, the attention scores of consecutive $M - r$ queries can be covered with a $M \times M$ block matrix, which can be computed using our oracle. Transformers with attention sink is similar to transformers with sliding window, except that we have an extra "sink window" in the beginning for all the queries. These sinks windows can be computed using $O(\frac{N}{M})$ oracle calls.

**Theorem 5.1.** *For any transformer $\mathcal{T}$ with $L$ layers, $H$ attention heads in each layer, input length $N$, embedding dimension $d$, constant-size sliding window, there exists an algorithm that simulates $\mathcal{T}$ with $O(\frac{N}{M} \cdot \frac{HL}{H'L'})$ calls to a transformer oracle with $L'$ layers, $H'$ attention heads in each layer, input length $M$ and embedding dimension $O(\frac{dH'L'}{H})$ with causal masking. This result still holds if we have constant-size attention sink.*

# 6    Acknowledgments

We thank Daniel Hsu, Jingwen Liu and Vatsal Sharan for useful discussions.

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

# A    Transformers as Basic Functions

We show that a single layer, single head transformer (attention mechanism with two MLPs) can act as a few different basic functions on the input matrix $X$ that we will need later.

First we show that we can turn any matrix $X$ into a matrix that is consisted of a block matrix of ones and zeros everywhere else.

**Lemma A.1.** *There exists a fixed matrix $U \in \mathbb{R}^{(d+1) \times d}$ and input MLP $\phi : \mathbb{R}^d \to \mathbb{R}^{d+1}$ such that for any input $X \in \mathbb{R}^{N \times d}$,*

$$\phi(X)U = \begin{pmatrix} \mathbf{1}_{a \times b} & 0 \\ 0 & 0 \end{pmatrix}$$

*for any $a \leq N, b \leq d$.*

*Proof.* Let $x_i = X[i, :]$ for all $i \in [N]$. Define $\phi(x_i) = (x_i, 1)$ if $i \leq a$ and $\phi(x_i) = (x_i, 0)$ if $i \geq a + 1$. Let

$$U = \begin{pmatrix} 0_{d \times b} & 0_{d \times (d-b)} \\ 1_{1 \times b} & 0_{d \times (d-b)} \end{pmatrix}$$

It is straightforward to check that $\phi(X)U$ is the matrix desired.    $\square$

Using this, we can construct a transformer that computes the sum of all input tokens.

**Lemma A.2.** *There exists a single layer, single head transformer with embedding dimension $d$ such that, on input matrix $X \in \mathbb{R}^{N \times d}$, it computes $\sum_{i=1}^{N} X[i, :] \in \mathbb{R}^{1 \times d}$.*

*Proof.* By Lemma A.1, there exists $W^Q, W^K$ such that $(XW^Q)[i, :] = (XW^K)[i, :] = \mathbf{1}_{1 \times d}$ and $k_i = N \cdot X[i, :]$ for all $1 \leq i \leq N$. As a result, the output for any token will exactly be $\sum_{i=1}^{N} X[i, :]$.    $\square$

A single layer, single head transformer also allows us to construct a look-up table such that each token can find information from other tokens.

**Lemma A.3** (Lemma D.1 of [SHT24]). *Given input matrix $X \in \mathbb{R}^{N \times d}$, an indexing function $\tau : \mathbb{R}^d \times [N] \to [N]$ and $f : \mathbb{R}^m \to \mathbb{R}^m$, there exists a single layer, single head transformer with embedding dimension $d$ such that the $i$-th output is $\rho(X[\tau(X[i, :], i), :])$.*

# B    Missing Proofs in Section 3

**Lemma B.1** (Lemma 3.1). *For any single layer, single head transformer $\mathcal{T}$ with input length $N$, embedding dimension $d$, there exists an algorithm that simulates $\mathcal{T}$ with $O(\frac{N^2}{M^2})$ calls to a transformer oracle with input length $M$ and embedding dimension $O(d)$.*

*Proof.* Let $\mathcal{T}$ be any single layer, single head transformer with query, key, value matrices $W^Q, W^K, W^V \in \mathbb{R}^{d \times d}$ with arbitrary input $X \in \mathbb{R}^{N \times d}$. Let $B$ be an upper bound of the absolute value of all entries in $X, W^Q, W^K, W^V$. Without loss of generality we can assume the first MLP $\phi$ is the identity function because otherwise we will compose it with the input MLP in the oracles. In addition, we can also assume that the layer MLP is the identity function, and this is because if not, we can first set the layer MLP in our oracle to be the identity function to compute the output of the large transformer before the layer MLP, and then use $O(\frac{N}{M})$ oracles as layer MLP to compute the final output.

Define (as usual) $Q = XW^Q, K = XW^K, V = XW^V$ and $q_i = Q[i, :], k_i = K[i, :], v_i = V[i, :]$ for all $1 \leq i \leq N$, and let $S_t = \{(t-1)M, \ldots, tM\}$ for all $1 \leq t \leq \frac{N}{M}$. Our goal is to simulate

$$\text{softmax}(q_i K^\top) V = \frac{\sum_{j=1}^{N} \exp(\langle q_i, k_j \rangle) \cdot v_j}{\sum_{j=1}^{N} \exp(\langle q_i, k_j \rangle)} =: \frac{\sum_{j=1}^{N} b_{i,j}}{\sum_{j=1}^{N} a_{i,j}} = \frac{\sum_{t=1}^{N/M} B_{i,t}}{\sum_{t=1}^{N/M} A_{i,t}} \tag{1}$$

for all $1 \leq i \leq N$, where we define

$$a_{i,j} := \exp(\langle q_i, k_j \rangle), b_{i,j} := \exp(\langle q_i, k_j \rangle) \cdot v_j.$$

and $A_{i,t} := \sum_{j \in S_t} a_{i,j}$ and $B_{i,t} = \sum_{j \in S_t} b_{i,j}$. Our algorithm can be summarized as the following two steps:

1. (Step 1) We calculate $A_{i,t}$ for all $1 \leq i \leq N, 1 \leq t \leq N/M$ using $(\frac{N}{M})^2$ oracle calls.

2. (Step 2) For each $t$, we exactly compute $\frac{B_{i,t}}{A_{i,t}}$ for all $i \in [N]$ using $(\frac{N}{M})^2$ oracle calls. Since we already know $A_{i,t}$ for all $i, t$, we can now compute

$$\frac{\sum_{t=1}^{N/M} B_{i,t}}{\sum_{t=1}^{N/M} A_{i,t}} = \frac{\sum_{t=1}^{N/M} \frac{B_{i,t}}{A_{i,t}} \cdot A_{i,t}}{\sum_{t=1}^{N/M} A_{i,t}}$$

for all $i$. Notice that this can be done either trivially or with $O((\frac{N}{M})^2)$ oracle calls with Lemma A.2 and the fact that we allow MLPs to do division.

**Step 1: Calculating $A_{i,t}$ for all $i, t$.** We first consider the case when $i \in S_t$. For any $1 \leq t \leq \frac{N}{M}$, define

$$W^{Q'} = \begin{pmatrix} W^Q & 0 \\ 0 & 1 \end{pmatrix}, W^{K'} = \begin{pmatrix} W^K & 0 \\ 0 & 1 \end{pmatrix} \in \mathbb{R}^{(d+1) \times (d+1)}$$

and MLP $\phi$ such that

$$Q' := \phi(X[S_t, :]) \cdot W^{Q'} = \begin{pmatrix} X[S_t, :] & 1_{M \times 1} \\ 0_{1 \times d} & 1 \end{pmatrix} \cdot \begin{pmatrix} W^Q & 0_{d \times 1} \\ 0_{1 \times d} & 1 \end{pmatrix} = \begin{pmatrix} q_{(t-1)M+1} & 1 \\ \vdots & \vdots \\ q_{tM} & 1 \\ 0 & 1 \end{pmatrix},$$

$$K' = \phi(X[S_t, :]) \cdot W^{K'} = \begin{pmatrix} X[S_t, :] & 1_{M \times 1} \\ 0_{1 \times d} & 1 \end{pmatrix} \cdot \begin{pmatrix} W^K & 0_{d \times 1} \\ 0_{1 \times d} & 1 \end{pmatrix} = \begin{pmatrix} k_{(t-1)M+1} & 1 \\ \vdots & \vdots \\ k_{tM} & 1 \\ 0 & 1 \end{pmatrix},$$

$$V' = \begin{pmatrix} 1_{M \times d} & 0_{M \times 1} \\ 0_{1 \times d} & 0 \end{pmatrix}.$$

(We can construct $W^{V'}$ and add an extra dimension using MLP such that we obtain $V'$ using Lemma A.1, but we omit it for simplicity. In particular, we will only need to use the first column of $V'$, so for what follows below, $V'$ could be $(1, \ldots, 1, 0)^\top \in \mathbb{R}^{(M+1) \times 1}$. We keep our notation consistent and let it have $d + 1$ columns.) Therefore, we have

$$Q'(K')^\top = \begin{pmatrix} \langle q_{(t-1)M+1}, k_{(t-1)M+1} \rangle + 1 & \cdots & \langle q_{(t-1)M+1}, k_{tM} \rangle + 1 & 1 \\ \vdots & \ddots & \vdots & \vdots \\ \langle q_{tM}, k_{(t-1)M+1} \rangle + 1 & \cdots & \langle q_{tM}, k_{tM} \rangle + 1 & 1 \\ 1 & \cdots & 1 & 1 \end{pmatrix}.$$

Finally, we can calculate that the entries of $\mathrm{softmax}(Q'(K')^\top)V'$ are given by

$$\frac{\sum_{j \in S_t} \exp(\langle q_i, k_j \rangle + 1)}{\sum_{j \in S_t} \exp(\langle q_i, k_j \rangle + 1) + \exp(1)} = \frac{\sum_{j \in S_t} a_{i,j}}{\sum_{j \in S_t} a_{i,j} + \exp(0)} = \frac{A_{i,t}}{A_{i,t} + \exp(0)}$$

for all $i \in S_t$. Therefore, we can use the MLP layer to calculate $A_{i,t}$ as:

$$A_{i,t} = \frac{\exp(0) \cdot \frac{A_{i,t}}{A_{i,t} + \exp(0)}}{1 - \frac{A_{i,t}}{A_{i,t} + \exp(0)}}.$$

Now we compute $A_{i,t}$ when $i \in S_{t'}$ for some $t' \neq t$. The high-level idea is similar, but now we feed the oracle with $[X[S_t, :], X[S_{t'}, :]] \in \mathbb{R}^{M \times 2d}$ and we let the MLP $\phi$ be such that

$$\phi([X[S_t, :], X[S_{t'}, :]]) = \begin{pmatrix} X[S_t, :] & X[S_{t'}, :] & 1 \\ 0_{1 \times d} & 0_{1 \times d} & 1 \end{pmatrix} \in \mathbb{R}^{(M+1) \times (2d+1)}$$

and we furthermore define

$$W^{Q''} = \begin{pmatrix} 0_{d\times d} & 0_{d\times 1} \\ W^Q & 0_{d\times 1} \\ 0_{1\times d} & 1_{1\times d} \end{pmatrix}, W^{K''} = \begin{pmatrix} W^K & 0_{d\times 1} \\ 0_{d\times d} & 0_{d\times 1} \\ 0_{1\times d} & 1_{1\times d} \end{pmatrix} \in \mathbb{R}^{(2d+1)\times(d+1)}$$

such that

$$Q'' := \phi([X[S_t,:], X[S_{t'},:]]) \cdot W^{Q''} = \begin{pmatrix} q_{(t'-1)M+1} & 1 \\ \vdots & \vdots \\ q_{t'M} & 1 \\ 0_{1\times d} & 1 \end{pmatrix} \in \mathbb{R}^{(M+1)\times(d+1)}$$

$$K'' := \phi([X[S_t,:], X[S_{t'},:]]) \cdot W^{K''} = \begin{pmatrix} k_{(t-1)M+1} & 1 \\ \vdots & \vdots \\ k_{tM} & 1 \\ 0_{1\times d} & 1 \end{pmatrix} \in \mathbb{R}^{(M+1)\times(d+1)}$$

$$V'' := \begin{pmatrix} 1_{M\times d} & 0_{M\times 1} \\ 0_{1\times d} & 0 \end{pmatrix} \in \mathbb{R}^{(M+1)\times(d+1)}.$$

Therefore, we have

$$Q''(K'')^\top = \begin{pmatrix} \langle q_{(t'-1)M+1}, k_{(t-1)M+1}\rangle + 1 & \cdots & \langle q_{(t'-1)M+1}, k_{tM}\rangle + 1 & 1 \\ \vdots & \ddots & \vdots & \vdots \\ \langle q_{t'M}, k_{(t-1)M+1}\rangle + 1 & \cdots & \langle q_{t'M}, k_{tM}\rangle + 1 & 1 \\ 1 & \cdots & 1 & 1 \end{pmatrix}.$$

We can now compute $A_{i,t}$ for all $i \in S_{t'}$ for the exact same reason as above. Step 1 requires $(\frac{N}{M})^2$ oracle calls.

**Step 2: Computing $\frac{B_{i,t}}{A_{i,t}}$ for all $i, t$.** If $i \in S_t$, we can compute $\frac{B_{i,t}}{A_{i,t}}$ using one oracle call simply by feeding the oracle with $X[S_t,:], W^Q, W^K, W^V$. It remains to compute $\frac{B_{i,t}}{A_{i,t}}$ for all $i \in S_{t'}$ where $t' \neq t$. We feed the oracle with $[X[S_t,:], X[S_{t'},:]]$ and let

$$W^{Q'''} = \begin{pmatrix} 0_{d\times d} \\ W^Q \end{pmatrix}, W^{K'''} = \begin{pmatrix} W^K \\ 0_{d\times d} \end{pmatrix}, W^{V'''} = \begin{pmatrix} W^V \\ 0_{d\times d} \end{pmatrix} \in \mathbb{R}^{2d\times d}$$

such that

$$Q''' := [X[S_t,:], X[S_{t'},:]] \cdot W^{Q'''} = \begin{pmatrix} q_{(t'-1)M+1} \\ \vdots \\ q_{t'M} \end{pmatrix},$$

$$K''' := [X[S_t,:], X[S_{t'},:]] \cdot W^{K'''} = \begin{pmatrix} k_{(t-1)M+1} \\ \vdots \\ k_{tM} \end{pmatrix},$$

$$V''' := [X[S_t,:], X[S_{t'},:]] \cdot W^{V'''} = \begin{pmatrix} v_{(t-1)M+1} \\ \vdots \\ v_{tM} \end{pmatrix}.$$

A simple calculation shows that $\text{softmax}(Q'''(K''')^\top)V'''$ gives us $\frac{B_{i,t}}{A_{i,t}}$ for $i \in S_{t'}$. In total we need $\frac{N}{M}$ oracle calls in step 2. $\qquad\square$

**Corollary B.2** (Corollary 3.2). *For any transformer $\mathcal{T}$ with $L$ layers, $H$ attention heads in each layer, input length $N$, embedding dimension $d$, there exists an algorithm that simulates $\mathcal{T}$ with $O((\frac{N}{M})^2 \cdot HL)$ calls to a single head, single layer transformer oracle with input length $M$ and embedding dimension $O(\frac{d}{H})$.*

*Proof.* We simply compute $\mathcal{T}$ layer by layer and head by head. For each layer, we compute the output of each attention head, which requires $O((\frac{N}{M})^2 \cdot H)$ oracle calls using Lemma B.1, concatenate and repeat for each layer. $\qquad\square$

**Lemma B.3** (Lemma 3.3). *One transformer with $L'$ layers, $H'$ attention heads in each layer, input length $M$ and embedding dimension $O(\frac{dH'L'}{H})$ can be used to simulate $H'L'$ independent instances of single layer, single head transformers with input length $M$ and embedding dimension $\frac{d}{H}$.*

*Proof.* Given $H'L'$ independent instances, we label the instances by $(h, \ell) \in H' \times L'$ and concatenate them together by columns to $X \in \mathbb{R}^{M \times \frac{dH'L'}{H}}$ such that

$$X = [X_{1,1}, X_{1,2}, \ldots, X_{1,L'}, X_{2,1}, \ldots, X_{H',L'}],$$

where $X_{h,\ell}$ is the input of the $(h, \ell)$-th instance for $h \in [H'], \ell \in [L']$. As a result, in the first layer, $[X_{h,1}, \ldots, X_{h,L'}] \in \mathbb{R}^{M \times \frac{dL'}{H}}$ is sent to attention head $h$. Let $W^Q, W^K, W^V \in \mathbb{R}^{\frac{d}{H} \times \frac{d}{H}}$ denote the query, key and value matrices in the $(h, 1)$-th instance. We construct $W^{Q'}, W^{K'}, W^{V'}$ as

$$W^{Q'} = \begin{pmatrix} W^Q \\ 0_{\frac{d}{H} \times \frac{d}{H}} \\ \vdots \\ 0_{\frac{d}{H} \times \frac{d}{H}} \end{pmatrix}, W^{K'} = \begin{pmatrix} W^K \\ 0_{\frac{d}{H} \times \frac{d}{H}} \\ \vdots \\ 0_{\frac{d}{H} \times \frac{d}{H}} \end{pmatrix}, W^{V'} = \begin{pmatrix} W^V \\ 0_{\frac{d}{H} \times \frac{d}{H}} \\ \vdots \\ 0_{\frac{d}{H} \times \frac{d}{H}} \end{pmatrix} \in \mathbb{R}^{(dL'/H) \times d/H}$$

and layer MLP the same as the layer MLP in instance $(h, 1)$ such that attention head $h$ in layer 1 exactly computes the $(h, 1)$-th instance. Also observe that we are not performing any computation regarding $(h, \ell)$-th instance for any $\ell \neq 1$. The same argument holds for all $h$, and therefore layer one of the large transformer computes $(h, 1)$-th instance for all $1 \leq h \leq H'$. Since the outputs of all $H'$ attention heads are stored at predefined locations after each layer, we can repeat this process for $L'$ times to compute all instances. $\qquad\square$

**Theorem B.4** (Theorem 3.4). *For any transformer $\mathcal{T}$ with $L$ layers, $H$ attention heads in each layer, input length $N$, embedding dimension $d$, there exists an algorithm that simulates $\mathcal{T}$ with $O((\frac{N}{M})^2 \cdot \frac{HL}{H'L'})$ calls to a transformer oracle with $L'$ layers, $H'$ attention heads in each layer, input length $M$, embedding dimension $O(\frac{dH'L'}{H})$.*

*Proof.* This follows from Corollary B.2 and Lemma B.3. $\qquad\square$

**Theorem B.5** (Theorem 3.5). *For any transformer $\mathcal{T}$ with $L$ layers, $H$ attention heads in each layer, input length $N$, embedding dimension $d$ and causal masking, there exists an algorithm that simulates $\mathcal{T}$ with $O((\frac{N}{M})^2 \cdot \frac{HL}{H'L'})$ calls to a transformer oracle with $L'$ layers, $H'$ attention heads in each layer, input length $M$, embedding dimension $O(\frac{dH'L'}{H})$ and causal masking.*

The high-level idea is identical to Theorem B.4. We first show that a single layer, single head transformer with input length $M$ and causal masking can compute $\sum_{j=1}^{i} \exp(\langle q_i, k_j \rangle)$ for all $1 \leq i \leq M$ given input and all parameters.

**Claim B.6.** *Given any $X \in \mathbb{R}^{M \times d}, W^Q, W^K, W^V \in \mathbb{R}^{d \times d}$, one calls to a single layer, single head transformer oracle $\mathcal{O}$ with input length $M$, embedding dimension $O(d)$ and causal masking suffices to compute $\sum_{j=1}^{i} \exp(\langle q_i, k_j \rangle)$ for all $1 \leq i \leq M$.*

*Proof.* We define

$$W^{Q'} = \begin{pmatrix} W^Q & 0_{1 \times d} \\ 0_{d \times 1} & 1 \end{pmatrix}, W^{K'} = \begin{pmatrix} W^K & 0_{1 \times d} \\ 0_{d \times 1} & 1 \end{pmatrix} \in \mathbb{R}^{(d+1) \times (d+1)}$$

such that

$$Q' := \phi(X) \cdot W^{Q'} = \begin{pmatrix} 0_{1\times d} & 1 \\ X & 1_{M\times 1} \end{pmatrix} \cdot \begin{pmatrix} W^Q & 0_{1\times d} \\ 0_{d\times 1} & 1 \end{pmatrix} = \begin{pmatrix} 0 & 1 \\ q_1 & 1 \\ \vdots & \vdots \\ q_M & 1 \end{pmatrix} \in \mathbb{R}^{(M+1)\times(d+1)}$$

$$K' = \phi(X) \cdot W^{K'} = \begin{pmatrix} 0_{1\times d} & 1 \\ X & 1_{M\times 1} \end{pmatrix} \cdot \begin{pmatrix} W^K & 0_{1\times d} \\ 0_{d\times 1} & 1 \end{pmatrix} = \begin{pmatrix} 0 & 1 \\ k_1 & 1 \\ \vdots & \vdots \\ k_M & 1 \end{pmatrix} \in \mathbb{R}^{(M+1)\times(d+1)}$$

$$V' = \begin{pmatrix} 0 & 0_{1\times d} \\ 0_{M\times 1} & 1_{M\times d} \end{pmatrix} \in \mathbb{R}^{(M+1)\times(d+1)}.$$

As a result, we have

$$Q'(K')^\top = \begin{pmatrix} 1 & 1 & \cdots & 1 \\ 1 & \langle q_1, k_1 \rangle & \cdots & \langle q_1, k_M \rangle \\ \vdots & \vdots & \ddots & \vdots \\ 1 & \langle q_M, k_1 \rangle & \cdots & \langle q_M, k_M \rangle \end{pmatrix}.$$

Finally, we can calculate that the $(i,j)$-th entry $(2 \leq i \leq M+1)$ of the oracle output, $\text{softmax}(\text{mask}(Q'(K')^\top))V'$, is

$$\frac{\sum_{j=1}^{i-1} \exp(\langle q_{i-1}, k_j \rangle + 1)}{\sum_{j=1}^{i-1} \exp(\langle q_{i-1}, k_j \rangle + 1) + \exp(1)} = \frac{\sum_{j=1}^{i-1} a_{i-1,j}}{\sum_{j=1}^{i} a_{i-1,j} + \exp(0)}.$$

Finally, we can define the MLP such that it outputs

$$\frac{\exp(0) \cdot \frac{\sum_{j=1}^{i-1} a_{i-1,j}}{\sum_{j=1}^{i} a_{i-1,j} + \exp(0)}}{1 - \frac{\sum_{j=1}^{i-1} a_{i-1,j}}{\sum_{j=1}^{i} a_{i-1,j} + \exp(0)}} = \sum_{j=1}^{i-1} a_{i-1,j}$$

for all $2 \leq i \leq M+1$. $\square$

*Proof of Theorem B.5.* The proof is similar to the proof of Theorem B.4. First notice that Lemma B.3 still holds if we add causal masking to the transformers because the proof is not affected by causal masking. In addition, the analog of Corollary B.2 will hold even if we add causal masking to the transformers if we can show that Lemma B.1 holds under causal masking since the proof is the same. Therefore, it suffices to prove Lemma B.1 under causal masking.

Let $\mathcal{T}$ be any single layer, single head transformer with query, key, value matrices $W^Q, W^K, W^V \in \mathbb{R}^{d\times d}$ and causal masking, with arbitrary input $X \in \mathbb{R}^{N\times d}$. For the same reason as Lemma B.1, we assume without loss of generality that both the input MLP and layer MLP are identity functions, and we use the same notation as in Lemma B.1. Our goal is to approximate

$$\frac{\sum_{j=1}^{i} \exp(\langle q_i, k_j \rangle) \cdot v_j}{\sum_{j=1}^{i} \exp(\langle q_i, k_j \rangle)} =: \frac{\sum_{j=1}^{i} b_{i,j}}{\sum_{j=1}^{i} a_{i,j}}.$$

for all $1 \leq i \leq N$. Notice that we already include causal masking in this expression by only summing over $j \leq i$. Define

$$A_{i,t}^{(1)} := \sum_{j\in S_t \cap [i]} a_{i,j}, A_{i,t}^{(2)} := \sum_{j\in S_t - [i]} a_{i,j} \Rightarrow A_{i,t} = A_{i,t}^{(1)} + A_{i,t}^{(2)}$$

and

$$B_{i,t}^{(1)} := \sum_{j\in S_t \cap [i]} b_{i,j}, B_{i,t}^{(2)} := \sum_{j\in S_t - [i]} b_{i,j} \Rightarrow B_{i,t} = B_{i,t}^{(1)} + B_{i,t}^{(2)}.$$

For each $i \in S_t$, we want to compute

$$\frac{\sum_{t'=1}^{t-1} B_{i,t'} + B_{i,t}^{(1)}}{\sum_{t'=1}^{t-1} A_{i,t'} + A_{i,t}^{(1)}} = \frac{\sum_{t'=1}^{t-1} \left( \frac{B_{i,t'}^{(1)}}{A_{i,t'}^{(1)}} \cdot A_{i,t'}^{(1)} + \frac{B_{i,t'}^{(2)}}{A_{i,t'}^{(2)}} \cdot A_{i,t'}^{(2)} \right) + \frac{B_{i,t}^{(1)}}{A_{i,t}^{(1)}} \cdot A_{i,t}^{(1)}}{\sum_{t'=1}^{t-1} (A_{i,t'}^{(1)} + A_{i,t'}^{(2)}) + A_{i,t}^{(1)}}.$$

Just like Lemma B.1, we will compute $A_{i,t'}^{(1)}$, $A_{i,t'}^{(2)}$, $\frac{B_{i,t'}^{(1)}}{A_{i,t'}^{(1)}}$, $\frac{B_{i,t'}^{(2)}}{A_{i,t'}^{(2)}}$ with our oracle independently for all $1 \le t' \le t-1$ (each with one oracle call). In addition, $A_{i,t}^{(1)}$ can be computed with a single oracle call using Claim B.6, and $\frac{B_{i,t}^{(1)}}{A_{i,t}^{(1)}}$ can be computed using a single oracle call by simply feeding the oracle with $X[S_t, :], W^Q, W^K, W^V$.

To compute $A_{i,t'}^{(1)}$ and $\frac{B_{i,t'}^{(1)}}{A_{i,t'}^{(1)}}$ with one oracle each for all $i, t'$, we can use the exact same construction in step 1 in the proof of Lemma B.1 (this works because our oracle also has causal masking). To compute $A_{i,t'}^{(2)}$ and $\frac{B_{i,t'}^{(2)}}{A_{i,t'}^{(2)}}$, we first define $\phi$ such that

$$
\phi(X[S_t, :], X[S_{t'}, :]) = \begin{pmatrix} x_{tM} & 0_{1 \times d} \\ x_{tM-1} & x_{t'M} \\ \vdots & \vdots \\ x_{(t-1)M+1} & x_{(t'-1)M+2} \\ 0_{1 \times d} & x_{(t'-1)M+1} \end{pmatrix}.
$$

Notice that this is valid because $\phi$ has information on the position of all the tokens. Furthermore, Lemma A.3 allows us to use our oracle to perform this computation as well. We also let

$$
W^{Q'} = \begin{pmatrix} W^Q \\ 0_{d \times d} \end{pmatrix}, W^{K'} = \begin{pmatrix} 0^{d \times d} \\ W^K \end{pmatrix}
$$

such that

$$
Q' := \phi(X[S_t, :], X[S_{t'}, :]) \cdot W^{Q'} = \begin{pmatrix} q_{tM} \\ q_{tM-1} \\ \vdots \\ q_{(t-1)M+1} \\ 0_{1 \times d} \end{pmatrix},
$$

$$
K' := \phi(X[S_t, :], X[S_{t'}, :]) \cdot W^{K'} = \begin{pmatrix} 0_{1 \times d} \\ k_{t'M} \\ k_{t'M-1} \\ \vdots \\ k_{(t'-1)M+1} \end{pmatrix},
$$

$$
V' := \begin{pmatrix} 0_{1 \times d} & 0 \\ 1_{M \times d} & 0_{M \times 1} \end{pmatrix}.
$$

Now notice that

$$
Q'(K')^\top = \begin{pmatrix} 0 & \langle q_{tM}, k_{t'M} \rangle & \cdots & \langle q_{tM}, k_{(t'-1)M+1} \rangle \\ 0 & \langle q_{tM-1}, k_{t'M} \rangle & \cdots & \langle q_{tM-1}, k_{(t'-1)M+1} \rangle \\ \vdots & \vdots & \ddots & \vdots \\ 0 & \langle q_{(t-1)M+1}, k_{t'M} \rangle & \cdots & \langle q_{(t-1)M+1}, k_{(t'-1)M+1} \rangle \\ 0 & 0 & \cdots & 0 \end{pmatrix}
$$

Therefore, the $(tM - i + 1)$-th row of $\mathrm{softmax}(\mathrm{mask}(Q'(K')^\top))V'$, which corresponds to $q_i$, is computing

$$
\frac{\sum_{j=i+1}^{t'M} \exp(\langle q_i, k_j \rangle)}{\sum_{j=i+1}^{t'M} \exp(\langle q_i, k_j \rangle) + \exp(0)},
$$

which allows us to compute $\sum_{j=i+1}^{t'M} \exp(\langle q_i, k_j \rangle) = A_{i,t'}^{(2)}$ exactly. Furthermore, $\frac{B_{i,t'}^{(2)}}{A_{i,t'}^{(2)}}$ can be computed by letting

$$
W^{Q''} = \begin{pmatrix} W^Q \\ 0_{d \times d} \end{pmatrix}, W^{K''} = \begin{pmatrix} 0^{d \times d} \\ W^K \end{pmatrix}, W^{V''} = \begin{pmatrix} 0^{d \times d} \\ W^V \end{pmatrix}
$$

such that

$$Q'' := \phi(X[S_t,:], X[S_{t'},:]) \cdot W^{Q''} = \begin{pmatrix} q_{tM} \\ q_{tM-1} \\ \vdots \\ q_{(t-1)M+1} \end{pmatrix},$$

$$K'' := \phi(X[S_t,:], X[S_{t'},:]) \cdot W^{K''} = \begin{pmatrix} k_{t'M} \\ k_{t'M-1} \\ \vdots \\ k_{(t'-1)M+1} \end{pmatrix},$$

$$V'' := \begin{pmatrix} v_{t'M} \\ v_{t'M-1} \\ \vdots \\ v_{(t'-1)M+1} \end{pmatrix}.$$

In total we need $O((\frac{N}{M})^2)$ oracle calls, and the proof is complete. $\qquad\square$

## C  Missing Proofs in Section 4

### C.1  Missing Proofs in Section 4.1

**Theorem C.1** (Theorem 4.1)**.** *Let $\mathcal{T}$ be a transformer with $L$ layers, $H$ attention heads in each layer, input length $N$ and embedding dimension $d$. Suppose there exist absolute constants $C, D > 0$ such that*

$$\frac{1}{C} \leq a_{i,j} \leq C, \text{ and } DN \cdot \max_j \|b_{i,j}\|_2 \leq \Big\| \sum_{j=1}^{N} b_{i,j} \Big\|_2$$

*where*

$$a_{i,j} = \exp(\langle q_i, k_j \rangle), b_{i,j} = \exp(\langle q_i, k_j \rangle) \cdot v_j$$

*for any query $q_i, k_j, v_j$ in any attention head. There exists an algorithm using $O(\frac{N}{M} \cdot \frac{HL}{H'L'})$ oracle calls to a small transformer with $L'$ layers, $H'$ attention heads in each layer, embedding dimension $O(\frac{dH'L'}{H})$ to obtain an $(1 + \varepsilon)$ approximation of $\mathcal{T}$ with probability at least $0.9$ for any fixed constant $\varepsilon > 0$.*

*Proof.* By Corollary B.2 and Lemma B.3, it suffices to prove the Theorem with $H = L = 1$ because when we generalize, all the attention head computation will be in parallel with each other.

Let $\mathcal{T}$ be any attention head with query, key, value matrices $W^Q, W^K, W^V \in \mathbb{R}^{d \times m}$ with arbitrary input $X \in \mathbb{R}^{N \times d}$. Without loss of generality we can assume the first MLP $\phi$ is the identity function because otherwise we will compose our oracle MLP with $\phi$. In addition, we can also assume that the layer MLP is the identity function, and this is because if not, we can still use identity functions in our oracle layer MLPs to compute the output of the large transformer before the layer MLP, and then use $O(\frac{N}{M})$ oracles as MLP to compute the final output.

Define $Q = XW^Q, K = XW^K, V = XW^V$ such that $q_i = Q[i,:], k_i = K[i,:], v_i = V[i,:]$ for all $1 \leq i \leq N$, and let $S_t = \{(t-1)M, \ldots, tM\}$ for all $1 \leq t \leq \frac{N}{M}$. Our goal is to approximate

$$\text{softmax}(q_i K^\top)V = \frac{\sum_{j=1}^{N} \exp(\langle q_i, k_j \rangle) \cdot v_j}{\sum_{j=1}^{N} \exp(\langle q_i, k_j \rangle)} = \frac{\sum_{j=1}^{N} b_{i,j}}{\sum_{j=1}^{N} a_{i,j}} \tag{2}$$

for all $1 \leq i \leq N$. The algorithm will be divided into two parts like before.

**Step 1: Approximating $\sum_{j=1}^{N} a_{i,j}$ for all** $1 \leq i \leq N$**.** Let $i \in S_t$ for some $1 \leq t \leq \frac{N}{M}$. We pick a random permutation $\tau$ of $[N]$, and by Lemma A.3 we can use $O(N/M)$ oracle calls[1] (one for each

---

[1]In practice, one would likely perform this permutation directly rather than using oracle calls, such as using the pytorch utility `randperm`. However, since it is a negligible additional number of calls, we use oracle calls here to simplify the computational model.

$X[S_t,:])$ to map $x_i$ to $[x_i, x_{\tau(i)}]$ for all $1 \leq i \leq N$. Now we use the first MLP $\phi$ in the oracles to map $(X[S_t,:], X[\tau(S_t),:])$ to

$$\phi(X[S_t,:], X[\tau(S_t),:]) = \begin{pmatrix} X[S_t,:] & X[\tau(S_t),:] & 1 \\ 0_{1\times d} & 0_{1\times d} & 1 \end{pmatrix}$$

and we furthermore define

$$W^{Q'} = \begin{pmatrix} W^Q & 0_{d\times 1} \\ 0_{d\times d} & 0_{d\times 1} \\ 0_{1\times d} & 1 \end{pmatrix}, W^{K'} = \begin{pmatrix} 0_{d\times d} & 0_{d\times 1} \\ W^K & 0_{d\times 1} \\ 0_{1\times d} & 1 \end{pmatrix}$$

in the oracle such that

$$Q' := \phi(X[S_t,:], X[\tau(S_t),:])W^{Q'} = \begin{pmatrix} q_{(t-1)M+1} & 1 \\ \vdots & \vdots \\ q_{tM} & 1 \\ 0 & 1 \end{pmatrix} \in \mathbb{R}^{(M+1)\times(d+1)},$$

$$K' := \phi(X[S_t,:], X[\tau(S_t),:])W^{K'} = \begin{pmatrix} k_{\tau((t-1)M+1)} & 1 \\ \vdots & \vdots \\ k_{\tau(tM)} & 1 \\ 0 & 1 \end{pmatrix} \in \mathbb{R}^{(M+1)\times(d+1)},$$

$$V' := \begin{pmatrix} 1_{M\times d} & 0_{M\times 1} \\ 1_{d\times 1} & 1 \end{pmatrix}.$$

Now we can compute $\sum_{j\in S_t} a_{i,\tau(j)}$ for all $i$ using $\frac{N}{M}$ oracle calls (the remaining proof is exactly the same as the proof of Lemma 3.1). Our estimator of $\sum_{j=1}^{N} a_{i,j}$ will be

$$\frac{N}{M} \sum_{j\in S_t} a_{i,\tau(j)}.$$

Our estimator is unbiased because

$$\mathbf{E}\Big[\sum_{j\in S_t} a_{i,\tau(j)}\Big] = \frac{M}{N} \cdot \sum_{j=1}^{N} a_{i,j}.$$

We can use Hoeffding's Inequality to show that

$$\Pr\Big[\Big|\sum_{j\in S_t} a_{i,\tau(j)} \cdot \frac{N}{M} - \sum_{j=1}^{N} a_{i,j}\Big| \geq \frac{\varepsilon}{4}\sum_{j=1}^{N} a_{i,j}\Big] = \Pr\Big[\Big|\sum_{j\in S_t} a_{i,\tau(j)} - \frac{M}{N}\sum_{j=1}^{N} a_{i,j}\Big| \geq \frac{\varepsilon M}{4N}\sum_{j=1}^{N} a_{i,j}\Big]$$

$$\leq 2\exp\Big(-\frac{\frac{2\varepsilon^2 M^2}{16N^2} \cdot (\sum_{j=1}^{N} a_{i,j})^2}{M(C-\frac{1}{C})^2}\Big)$$

$$\leq 2\exp\Big(-\frac{2\varepsilon^2 M(N/C)^2}{16N^2C^2}\Big)$$

$$= 2\exp\Big(-\frac{2\varepsilon^2 M}{16C^4}\Big),$$

which is at most $\frac{1}{20N}$ if $M \geq \frac{8C^4\log(40N)}{\varepsilon^2}$, which is true by our assumption on $M$. A union bound over all $i \in [N]$ allows us to show that we get a $(1+\varepsilon/4)$ approximation of $\sum_{j=1}^{N} a_{i,j}$ for all $i$ with probability at least $0.95$.

**Step 2: Approximating** $\frac{\sum_{j=1}^{N} b_{i,j}}{\sum_{j=1}^{N} a_{i,j}}$ for all $1 \leq i \leq N$. Let $i \in S_t$ for some $1 \leq t \leq \frac{N}{M}$. We pick a random permutation $\tau$ of $[N]$, and define

$$\phi([X[S_t,:], X[\tau(S_t),:]]) = [X[S_t,:], X[\tau(S_t),:]]$$

and furthermore
$$W^{Q''} = \begin{pmatrix} W^Q \\ 0_{d\times d} \end{pmatrix}, W^{K''} = \begin{pmatrix} 0_{d\times d} \\ W^K \end{pmatrix}, W^{V''} = \begin{pmatrix} 0_{d\times d} \\ W^V \end{pmatrix}.$$

The output $\text{softmax}(Q''(K'')^\top)V''$ gives us exactly
$$\frac{\sum_{j\in S_t} b_{i,\tau(j)}}{\sum_{j\in S_j} a_{i,\tau(j)}},$$

which will be our estimator. First observe that
$$\mathbf{E}\Big[\sum_{j\in S_t} b_{i,\tau(j)}\Big] = \frac{M}{N}\cdot\sum_{j=1}^N b_{i,j}.$$

We can again use Hoeffding's Inquality to show that
$$\Pr\Big[\Big\|\frac{N}{M}\sum_{j\in S_t} b_{i,\tau(j)} - \sum_{j=1}^N b_{i,j}\Big\|_2 \geq \frac{\varepsilon}{4}\Big\|\sum_{j=1}^N b_{i,j}\Big\|_2\Big] \leq 2\cdot\exp\Big(-\frac{\varepsilon^2 M\cdot\|\sum_{j=1}^N b_{i,j}\|_2^2}{128N^2(\max_j\|b_{i,j}\|_2)^2}\Big)$$
$$\leq 2\cdot\exp\Big(-\frac{\varepsilon^2 MD^2}{128}\Big)$$

which is at most $\frac{1}{20N}$ when $M\geq\frac{128(\log d+\log 40+\log N)}{\varepsilon^2 D^2}$. We have shown that
$$\frac{N}{(1+\varepsilon/4)M} \leq \frac{\sum_{j=1}^N a_{i,j}}{\sum_{j\in S_t} a_{i,\tau(j)}} \leq \frac{N}{(1-\varepsilon/4)M},$$

and now we prove
$$\Big\|\sum_{j=1}^N b_{i,j} - \sum_{j\in S_t} b_{i,\tau(j)}\cdot\frac{\sum_{j=1}^N a_{i,j}}{\sum_{j\in S_t} a_{i,\tau(j)}}\Big\|_2 \leq \varepsilon\cdot\Big\|\sum_{j=1}^N b_{i,j}\Big\|_2,$$

which concludes the proof. Indeed, we first decompose
$$\Big\|\sum_{j=1}^N b_{i,j} - \sum_{j\in S_t} b_{i,\tau(j)}\cdot\frac{\sum_{j=1}^N a_{i,j}}{\sum_{j\in S_t} a_{i,\tau(j)}}\Big\|_2 = \Big\|\sum_{j=1}^N b_{i,j} - \sum_{j\in S_t} b_{i,\tau(j)}\cdot\frac{N}{M} + (1-\delta)\sum_{j\in S_t} b_{i,\tau(j)}\cdot\frac{N}{M}\Big\|_2$$
$$\leq \Big\|\sum_{j=1}^N b_{i,j} - \sum_{j\in S_t} b_{i,\tau(j)}\cdot\frac{N}{M}\Big\|_2 + (1-\delta)\Big\|\sum_{j\in S_t} b_{i,\tau(j)}\cdot\frac{N}{M}\Big\|_2,$$
$$\tag{3}$$

where
$$\frac{1}{1+\varepsilon/4} \leq \delta := \frac{M\cdot\sum_{j=1}^N a_{i,j}}{N\cdot\sum_{j\in S_t} a_{i,\tau(j)}} \leq \frac{1}{1-\varepsilon/4}.$$

Now we already know
$$\Big\|\sum_{j=1}^N b_{i,j} - \sum_{j\in S_t} b_{i,\tau(j)}\cdot\frac{N}{M}\Big\|_2 \leq \frac{\varepsilon}{4}\cdot\Big\|\sum_{j=1}^N b_{i,j}\Big\|_2,$$

and therefore Equation 3 can be further bounded by
$$\frac{\varepsilon}{4}\cdot\Big\|\sum_{j=1}^N b_{i,j}\Big\|_2 + (1-\delta)(1+\varepsilon/4)\Big\|\sum_{j=1}^N b_{i,j}\Big\|_2$$
$$\leq \frac{\varepsilon}{4}\cdot\Big\|\sum_{j=1}^N b_{i,j}\Big\|_2 + \frac{\varepsilon}{4+\varepsilon}(1+\varepsilon/4)\Big\|\sum_{j=1}^N b_{i,j}\Big\|_2$$
$$\leq \varepsilon\cdot\Big\|\sum_{j=1}^N b_{i,j}\Big\|_2,$$

which concludes the proof. $\qquad\square$

## C.2 Missing Proofs in Section 4.2

**Theorem C.2** (Theorem 4.2). *Given $\frac{N}{M}$ instances of single layer, single head transformers with input length $M$ and embedding dimension $d$, there exists an algorithm that simulates them with one call of a single layer, single head transformer with input length $O(N)$ and embedding dimension $O(d)$, along with $O(\frac{N}{M})$ many matrix multiplications of size $M \times d \times d$.*

*Proof.* We assume that the input/layer MLPs in all the instances are identity functions for the same reason as in Lemma B.1. Let $X_i \in \mathbb{R}^{M \times d}$ be the input, $W_i^Q, W_i^K, W_i^V \in \mathbb{R}^{d \times d}$ be the query, key and value matrices for each instance $1 \leq i \leq \frac{N}{M}$. Our goal is to calculate

$$\text{softmax}((X_i W_i^Q)(X_i W_i^K)^\top)(X_i W_i^V)$$

for all $1 \leq i \leq \frac{N}{M}$. We first compute $X_i W_i^Q, X_i W_i^K, X_i W_i^V$ for all $1 \leq i \leq \frac{N}{M}$. For convenience we will denote

$$q_{i,j} = (X_i W_i^Q)[j,:], k_{i,j} = (X_i W_i^K)[j,:], v_{i,j} = (X_i W_i^V)[j,:]$$

such that our goal is to compute $\text{softmax}(q_{i,j}(X_i W_i^K)^\top)(X_i W_i^V)$ for all $1 \leq i \leq \frac{N}{M}, 1 \leq j \leq M$. For convenience we let

$$\|X_i\|_{\infty,2} \cdot \|W_i^Q\|_2, \|X_i\|_{\infty,2} \cdot \|W_i^K\|_2, \|X_i\|_{\infty,2} \cdot \|W_i^V\|_2 \leq C \leq \text{poly}(N),$$

for some $C$ (because we assume that all entries have $O(\log N)$ bit representation, and therefore each parameter is at most $\text{poly}(N)$). As a result,

$$-C^2 \leq \langle q_{i,j}, k_{i',j'} \rangle \leq C^2, \|v_{i,j}\|_2 \leq C$$

for any $1 \leq i, i' \leq N/M, 1 \leq j, j' \leq M$.

Let $B \in \mathbb{R}$ to be determined. Define

$$u_1, \ldots, u_{\frac{N}{M}} \in \{0, -B\}^r$$

such that $\binom{r}{r/2} \geq N/M$ (we only need $r \leq O(\log(N/M))$ by Stirling approximation) and all $u_i$ have exactly $r/2$ zeros (if there are more vectors than needed, simply make sure they are distinct), and let

$$v_i = B \cdot (1, \ldots, 1) + u_i$$

for all $1 \leq i \leq \frac{N}{M}$ such that $\langle u_i, v_i \rangle = 0$ for all $i$. For any $i \neq j$, notice that there must exist an index at which $u_i$ is $-B$ and $v_j$ is $B$. This is because the set of nonzeros in $v_i$ is the compliment of the set of nonzeros in $u_i$, and the former set must be different from the set of nonzeros in $v_j$. Now append $u_i$ to $q_{i,j}$ to get $q'_{i,j}$ and $v_i$ to $k_{i,j}$ to get $k'_{i,j}$ for all $1 \leq j \leq M$. Set $d' = d + r$. For each pair of $q'_{i,j}$ and $k'_{i',j'}$:

- If $i = i'$, i.e. $q_i$ and $k_{i'}$ are in the same small transformer instance, then

$$\langle q'_{i,j}, k'_{i',j'} \rangle = \langle q_{i,j}, k_{i',j'} \rangle + \langle u_i, v_{i'} \rangle = \langle q_{i,j}, k_{i',j'} \rangle$$

- If $i \neq i'$, i.e. $q_i$ and $k_{i'}$ are in different small transformer instances, then

$$\langle q'_{i,j}, k'_{i',j'} \rangle = \langle q_{i,j}, k_{i',j'} \rangle + \langle u_i, v_{i'} \rangle \leq \langle q_{i,j}, k_{i',j'} \rangle - B^2.$$

Now we let $Q \in \mathbb{R}^{N \times d'}$ be the matrix of $q'_{i,j}$ and $K \in \mathbb{R}^{N \times d'}$ be the matrix of $k'_{i,j}$ and $V = X_i W_i^V$. We set $X = [Q, K, V] \in \mathbb{R}^{N \times (3d')}$ (note that we can always pad zeros to $V$ to make dimensions match),

$$W^Q = \begin{pmatrix} I_{d' \times d'} \\ 0_{d' \times d'} \\ 0_{d' \times d'} \end{pmatrix}, W^K = \begin{pmatrix} 0_{d' \times d'} \\ I_{d' \times d'} \\ 0_{d' \times d'} \end{pmatrix}, W^V = \begin{pmatrix} 0_{d' \times d'} \\ 0_{d' \times d'} \\ I_{d' \times d'} \end{pmatrix}$$

such that

$$[Q, K, V]W^Q = Q, [Q, K, V]W^K = K, [Q, K, V]W^V = V.$$

Furthermore, for each query $(q_{i,j}, u_i)$, its inner product between all keys in the same small transformer will be preserved, while its inner product between all keys in a different small transformer will be at most $C^2 - B^2$. Therefore, for each query $q_{i,j}$, we can calculate the error as:

$$\Big\| \sum_{j'=1}^{M} \frac{\exp(\langle q_{i,j}, k_{i,j'}\rangle)}{\sum_{\ell=1}^{M} \exp(\langle q_{i,j}, k_{i,\ell}\rangle)} \cdot v_{i,j'} - \sum_{i'=1}^{N/M} \sum_{j'=1}^{M} \frac{\exp(\langle q'_{i,j}, k'_{i',j'}\rangle)}{\sum_{\ell=1}^{N/M} \sum_{\ell'=1}^{M} \exp(\langle q'_{i,j}, k'_{\ell,\ell'}\rangle)} v_{i',j'} \Big\|_2$$

$$\leq \Big\| \sum_{j'=1}^{M} \frac{\exp(\langle q_{i,j}, k_{i,j'}\rangle)}{\sum_{\ell=1}^{M} \exp(\langle q_{i,j}, k_{i,\ell}\rangle)} \cdot v_{i,j'} - \sum_{j'=1}^{M} \frac{\exp(\langle q'_{i,j}, k'_{i,j'}\rangle)}{\sum_{\ell=1}^{N/M} \sum_{\ell'=1}^{M} \exp(\langle q'_{i,j}, k'_{\ell,\ell'}\rangle)} \cdot v_{i,j'} \Big\|_2$$

$$+ \frac{NC \exp(C^2 - B^2)}{\sum_{\ell=1}^{M} \exp(\langle q_{i,j}, k_{i,\ell}\rangle)}$$

$$\leq \Big\| \sum_{j'=1}^{M} \Big( \frac{\exp(\langle q_{i,j}, k_{i,j'}\rangle)}{\sum_{\ell=1}^{M} \exp(\langle q_{i,j}, k_{i,\ell}\rangle)} \cdot v_{i,j'} - \frac{\exp(\langle q_{i,j}, k_{i,j'}\rangle)}{\sum_{\ell=1}^{N/M} \sum_{\ell'=1}^{M} \exp(\langle q'_{i,j}, k'_{\ell,\ell'}\rangle)} \cdot v_{i,j'} \Big) \Big\|_2$$

$$+ \frac{NC \exp(C^2 - B^2)}{M \exp(-C^2)}.$$

Now notice that

$$\sum_{\ell=1}^{N/M} \sum_{\ell'=1}^{M} \exp(\langle q'_{i,j}, k'_{\ell,\ell'}\rangle) = \sum_{\ell=1}^{M} \exp(\langle q_{i,j}, k_{i,\ell}\rangle) + \sum_{\ell=1, \ell\neq i}^{N/M} \sum_{\ell'=1}^{M} \exp(\langle q'_{i,j}, k'_{\ell,\ell'}\rangle)$$

and that

$$M \exp(-C^2) \leq \sum_{\ell=1}^{M} \exp(\langle q_{i,j}, k_{i,\ell}\rangle) \leq M \exp(C^2)$$

$$N \exp(-C^2 - B^2) \leq \sum_{\ell=1, \ell\neq i}^{N/M} \sum_{\ell'=1}^{M} \exp(\langle q'_{i,j}, k'_{\ell,\ell'}\rangle) \leq N \exp(C^2 - B^2).$$

As a result,

$$\Big| \frac{1}{\sum_{\ell=1}^{M} \exp(\langle q_{i,j}, k_{i,\ell}\rangle)} - \frac{1}{\sum_{\ell=1}^{N/M} \sum_{\ell'=1}^{M} \exp(\langle q'_{i,j}, k'_{\ell,\ell'}\rangle)} \Big| \leq \frac{N \exp(C^2 - B^2)}{M^2 \exp(-C^2)} = \frac{N \exp(2C^2 - B^2)}{M^2}.$$

Therefore, we can further upper bound the error by

$$M \cdot \exp(C^2) \cdot C \cdot \frac{N \exp(2C^2 - B^2)}{M^2} + \frac{NC \exp(C^2 - B^2)}{M \exp(-C^2)}$$

$$= \frac{NC \exp(2C^2 - B^2)(1 + \exp(C^2))}{M}.$$

Therefore, we can set $B \leq \mathrm{poly}(N)$ to be sufficiently large such that the error is $O(\frac{1}{2^N})$. $\qquad\square$

## D  Missing Proofs in Section 5

**Theorem D.1** (Theorem 5.1). *For any transformer $\mathcal{T}$ with $L$ layers, $H$ attention heads in each layer, input length $N$, embedding dimension $d$, constant-size sliding window, there exists an algorithm that simulates $\mathcal{T}$ with $O(\frac{N}{M} \cdot \frac{HL}{H'L'})$ calls to a transformer oracle with $L'$ layers, $H'$ attention heads in each layer, input length $M$ and embedding dimension $O(\frac{dH'L'}{H})$ with causal masking. This result still holds if we have constant-size attention sink.*

*Proof.* The proof goes in the same way as Theorem B.5, where we assume without loss of generality that $H = L = H' = L' = 1$. We start with the sliding window scenario where the size of sliding window is $r$. Notice that for each query $q_i$ $(i \geq r + 1)$ we want to compute

$$\sum_{j=i-r+1}^{i} \exp(\langle q_i, k_j\rangle) = \sum_{j=1}^{i} \exp(\langle q_i, k_j\rangle) - \sum_{j=1}^{i-r} \exp(\langle q_i, k_j\rangle),$$

and
$$\sum_{j=i-r+1}^{i} \exp(\langle q_i, k_j \rangle) \cdot v_j = \sum_{j=1}^{i} \exp(\langle q_i, k_j \rangle) \cdot v_j - \sum_{j=1}^{i-r} \exp(\langle q_i, k_j \rangle) \cdot v_j$$

given window size $r$. In addition, we can compute $\sum_{j=1}^{r} \exp(\langle q_i, k_j \rangle)$ and $\sum_{j=1}^{r} \exp(\langle q_i, k_j \rangle) \cdot v_j$ for all $i \leq r$ with 2 oracle calls, as shown in the proof of Theorem B.5.

We will partition $\{r+1, \ldots, N\}$ into chunks $S_1' = \{r+1, \ldots, M\}$, $S_2' = \{M+1, \ldots, 2M-r\}, \ldots$ of size $M-r$ and approximate the terms above for all $i \in S_t'$ in each chunk using constant many oracle calls, which suffices since $r$ is a constant. Without loss of generality we only prove our claim for $S_1'$ as the proof can be easily generalized to the remaining $S_t'$. Indeed, observe that

$$\sum_{j=1}^{i} \exp(\langle q_i, k_j \rangle) \text{ and } \sum_{j=1}^{i-r} \exp(\langle q_i, k_j \rangle)$$

for all $i \in S_1'$ can both be computed exactly with one oracle call using the proof of Claim B.6. Furthermore, one oracle call suffices to compute either

$$\frac{\sum_{j=1}^{i} \exp(\langle q_i, k_j \rangle) \cdot v_j}{\sum_{j=1}^{i} \exp(\langle q_i, k_j \rangle)} \text{ or } \frac{\sum_{j=1}^{i-r} \exp(\langle q_i, k_j \rangle) \cdot v_j}{\sum_{j=1}^{i-r} \exp(\langle q_i, k_j \rangle)}$$

for all $i \in S_1'$ because we can feed the oracle with $X[S_1, :]$ and use $W^Q$ to project $X[S_1, :]$ to $\{q_{r+1}, \ldots, q_M\}$ and use $W^K$ to project $X[S_1, :]$ to $\{k_1, \ldots, k_{M-r}\}$ (similar to proof of Lemma B.1).

Finally, when there are attention sinks, we simply need to further calculate $\sum_{j=1}^{s} \exp(\langle q_i, k_j \rangle)$ and $\sum_{j=1}^{s} \exp(\langle q_i, k_j \rangle) \cdot v_j$. Observe that

$$\sum_{j=1}^{s} \exp(\langle q_i, k_j \rangle) = \sum_{j=1}^{i} \exp(\langle q_i, k_j \rangle) - \sum_{j=s+1}^{i} \exp(\langle q_i, k_j \rangle),$$

$$\sum_{j=1}^{s} \exp(\langle q_i, k_j \rangle) \cdot v_j = \sum_{j=1}^{i} \exp(\langle q_i, k_j \rangle) \cdot v_j - \sum_{j=s+1}^{i} \exp(\langle q_i, k_j \rangle) \cdot v_j.$$

Each of the four terms can be computed with the exact same technique as in sliding window with 1 oracle call. $\qquad\square$

