# OpenReview forum: "Two Heads are Better than One: Simulating Large Transformers with Small Ones"
_NeurIPS.cc/2025/Conference — NeurIPS 2025 spotlight_

### Official Review · Reviewer_TTT1 · 2025-06-22

**Clarity:** 2
**Significance:** 2
**Originality:** 3
**Rating:** 3
**Confidence:** 4

**Summary:**

This paper theoretically prove that large transformer (input length N) can be simulated by no more than $O((N/M)^2)$ small transformers (input length M, where M << N) with acceptable errors. Some practical scenarios like masking, sliding window and attention sink are also considered in this paper. This theoretical result may be useful for efficiently processing long sequences by leveraging hardware optimizations designed for smaller sequences.

**Questions:**

Q1: This paper provide a $(1+\epsilon)^2$ multiplicative error bound for a single layer. Will the errors explode when considering practical Transformer with multi-layer and other non-linear components like LayerNorm and  feed forward layer?

Q2: The Abstract claims that a large Transformer can be efficiently simulated by only $O((N/M)^2)$ small Transformers, and that this cannot be improved in the worst case.

How to understand "cannot be improved"?        Have the authors proved it?

Q3: Since this paper mentions that modern GPUs can accelerate small input sequences in transformers, it is recommended that the authors consider performing some experients to support the utility their algorithm.

**Ethical Concerns:**

["NO or VERY MINOR ethics concerns only"]

**Final Justification:**

I agree that your theoretical results are interesting, but I do not think the readers would be willing or easily follow your work without empirical foundation.

**Limitations:**

yes

**Paper Formatting Concerns:**

1. The title formatting does not follow standard academic capitalization conventions.

2. The abstract should be limited to one paragraph.

**Quality:**

1

**Strengths And Weaknesses:**

Strengths:

1. This paper proves that large transformer can be simulated by multiple small transformer, and establishs the guarrantee for the worst-case inputs: quadratic small transformers are sufficient and necessary.

2. The paper's simulating method preserves the representational strength of standard Transformers. The simulation error for a single layer is bounded by a $(1+\epsilon)^2$ multiplicative error bound.

3. The paper considers practical setting like causal masking, sliding window, and attention sinks.

Weaknesses:

1. Lack of experiments: While I believe that multiple small models can simulate a large model from the perspective of representational strength, the viability of the algorithm should be verified by experiments.

2. The writing is redundant: Definition 2.3 - 2.5 are very similar (page 6). Theorem 3.4 and Theorem 3.5 differ by only one word (line 337-346).

3. The theorems are informal: The theorems in this paper are mainly text-based and are not expressed mathematically, which affects readability.

Minors and Typos:

1. The title formatting does not follow standard academic capitalization conventions.

2. The abstract should be limited to one paragraph.

3. criticaly -> critically (line 79)

4. Vectors and matrices should be bolded

---

> ### Author Rebuttal · Authors · 2025-07-31
>
> We thank the reviewer for the comments. Here are our responses.
>
> **Empirical validation:** We agree that our paper gives rise to many interesting questions which should be the study of follow-up empirical work. However, there are two main reasons we chose not to include them here.
>
> First, we feel that the theory results presented in our paper stand on their own. The goal of our work here is to perform a complete theoretical analysis of how small transformers can be used to simulate big ones. It is important to nail down these precise dependencies before implementing the approach in practice, and that is exactly what we do here - we provide a complete and systematic analysis, including optimal and yet simple simulation algorithms. Most notably, some of our results show that surprisingly efficient simulations are possible (for example, Theorem 1.1 and Lemma 3.3 show that we can make use of all the layers of an oracle simultaneously), and we’re also able to rigorously pin down when techniques like attention sinks, sliding window, or assumptions on the data can yield real improvements.
>
> We also want to highlight that our results indicate that quadratic/linear amounts of small transformers have at least as much representational strength as a constant amount of large transformers in different settings. Such general representation results can only be proved theoretically.
>
> Second, an empirical validation of our results here that is truly meaningful would need to relate to real LLMs that are trained on large corpa of text. The implementation and parameter tuning can now build on our results, but this would be a substantial undertaking, and likely depends quite a bit on the setting like the reviewer suggests. Our simulation algorithms are simple and even “differentiable”, and therefore could be implemented effectively in practice.
>
> For these reasons, we feel that careful experimental work should comprise a second paper (or perhaps even more than one follow-up paper for different applications).
>
> **Writing is redundant; theorems are informal:** Although Definitions 2.3-2.5 are similar, we believe that it is important to distinguish causal masking, sliding window and attention sinks as they are all different in practice. In addition, these differences will affect our proof details, so we want to make sure that they are stated as clearly as possible; we need to formally define all the different terms and models, even if some are similar to each other, so that we can formally state and prove all our results. As for Theorem 3.4 and 3.5, we want to emphasize Theorem 3.4 as our main result whose proof is based on the lemmas in section 3, while Theorem 3.5 uses a different set of lemmas that are presented in appendix (supplementary material). The statements may appear similar, but they require substantially different work to prove.
>
> Our main result (Theorem 1.1) is stated formally in mathematical language. We use informal theorem (Theorem 1.2) for average case results because of the page limit, and we believe that this makes our results easier to understand in the introduction. All the precise, formally-stated theorems are presented and proved in the supplementary material.
>
> **Typos and formatting:** Thank you for the suggestions. We will adjust them accordingly.
>
> **Multiplicative error bound for single layer:** Our main result (Theorem 1.1) provides an exact algorithm if we assume infinite (full) bit precision. The reason we define simulating as an approximation up to $O(1/2^{N})$ multiplicative error is that, as in most theoretical works in this area, we assume only $O(\log n)$ bit precision to more accurately model realistic hardware, which means that all parameters are represented by $O(\log N)$ bits. In this case, it is impossible to achieve perfect accuracy because the $exp$ function in the attention mechanism will have to be approximated. In other words: this approximation error is unavoidable in limited-precision hardware. Our results provide the best possible approximation in this scenario.
>
> **Cannot be improved in the worst case:** Our main theorem (Theorem 1.1) states that $O((N/M)^2)$ small transformers suffice, and we discuss in line 109-113 that this is also optimal under standard complexity-theoretic assumptions. So yes we have proved it, but we did not make it a formal theorem because the argument only requires a few sentences.
>
> We hope our responses make sense, and we sincerely hope that the reviewer can reevaluate our work as we believe that our work has its important theoretical values. Please let us know if there are any further questions!

---

> > ### Comment · Reviewer_TTT1 · 2025-08-04
> >
> > Thank you for your responses.
> > I will keep my current score because no empirical evidence has been provided, and I do not think the theoretical result is significant.

---

> > > ### Author Response · Authors · 2025-08-04
> > >
> > > We thank the reviewer for the feedback. For what it's worth, we want to emphasize the two things:
> > > 1. Our paper establishes the theoretical foundations of simulating large transformers with smaller ones. Our algorithms are optimal and simple, and results involving representation can only be proved theoretically. Theoretical work such as ours is explicitly welcomed by the NeurIPS call for papers.
> > > 2. We believe that our theoretical work has its significant value as we explained in the paper and rebuttals. The reviews from other reviewers reaffirm its significance.

---

> > > > ### Comment · Reviewer_TTT1 · 2025-08-05
> > > >
> > > > Thank you for your response.
> > > >
> > > > In your rebuttal, you mention that your "paper gives rise to many interesting questions which **should** be the study of follow-up empirical work".
> > > >
> > > > I agree that your theoretical results are interesting, but I do not think the readers would be willing or easily follow your work without empirical foundation.
> > > >
> > > > I am willing to increase my score from Reject (2) to Borderline Reject (3), but I do not feel the paper meets the bar for acceptance at NeurIPS in its current form.

---

### Official Review · Reviewer_BQJT · 2025-06-26

**Clarity:** 4
**Significance:** 2
**Originality:** 3
**Rating:** 3
**Confidence:** 3

**Summary:**

This paper proves the existence of an algorithm to tightly approximate the output of a transformer with a context length of $N$ by making roughly $O((N/M)^2)$ calls to transformers with a shorter context length $M$, where $M \ll N$. While this algorithm still requires a quadratic number of operations relative to the context length, the authors suggest this decomposition could be leveraged to achieve wall-clock time speedups if modern GPUs/ASICs provide fast inference for moderate-to-short contexts. They also show that $O(N/M)$ calls to smaller transformers is optimal in streaming settings and in cases where key, query, and value elements are in relative close proximity.

**Questions:**

**Questions:**

1. It would be very helpful if you could provide more insight into when your method might yield the most gains. For example, for what context lengths ($N$) and model sizes does it provide the most benefits? Is it more helpful for prompt processing (pre-filling) or the inference stage?"

2. Is it fair to say the core of the proposed decomposition is Lemma 3.1, which decomposes a single large attention computation into multiple smaller ones? If so, how does this theoretical decomposition compare to implementations like FlashAttention, which also use similar tiling and softmax calculation tricks?

3. The actual algorithm is not provided in the paper, but proofs of existence/correctness are presented. Is it possible to present the algorithm itself, or at least provide more insight on it?

4.  Is the approximation error caused by the "$O(\log N)$-bit representations" assumption, or is the approximation a general part of the algorithm, even at full precision?

**Ethical Concerns:**

["NO or VERY MINOR ethics concerns only"]

**Final Justification:**

After carefully considering the authors' responses and the discussion among the other reviewers, I still believe the main weakness of the paper is its lack of empirical evidence to support the claimed performance gains. While I understand that this is presented as a theoretical paper, its premise and goal are fundamentally practical: to introduce a new method for handling long input sequences by leveraging modern GPUs. Therefore, the absence of any experimental results raises questions about its real-world applicability.

**Limitations:**

In my opinion, the lack of experimental results is not properly acknowledged in the limitations section: "our algorithms are efficient in theory, and we expect them to be efficient in practice as well"

**Quality:**

2

**Strengths And Weaknesses:**

**Strengths:**

The paper is very well-written and well-organized. I particularly appreciated that the authors provided informal versions of their main results in Section 1.2 and gave additional insights before diving into the full technical details. Also, the roadmap for the proof and the proof sketch of Lemma 3.1 are helpful for understanding the gist of the theory. Finally, the main result of the paper is very interesting on its own.


**Weaknesses:**

The main weakness of the paper is its lack of empirical evidence to support the claimed performance gains. I understand this is presented as a theory paper; however, its premise and goal is purely practical: to introduce a new method for handling long input sequences by leveraging modern GPUs. Therefore, the absence of experimental results makes its applicability questionable.

1. The method's applicability hinges on the paper's argument that "modern GPUs enable faster transformer inference with respect to the wall-clock time when the input sequence is short-to-moderate, then our algorithms allow faster wall-clock time inference." This critical, hardware-dependent condition is influenced by many factors, and whether it actually holds true needs to be shown experimentally.

2.  The number of calls to the oracle models can be very large. For instance, if $M \approx \sqrt{N}$, the method makes $O(N)$ calls to potentially $O(N)$ different models. Although the authors claim the "total number of parameters does not depend on $N$," it is unclear how the overhead of calling many (even slightly) different models compares to using one model over the whole context.

Determining if the proposed method provides any reasonable practical gain, and in what regime, depends highly on intricate hardware details (e.g., whether the pipeline is memory- or compute-bound), which can only be clarified through experimental results.

3.  The paper proves the existence of an algorithm that "approximates" the output of the large transformer. While the authors quantify the approximation error, it remains unclear how this error affects model performance compared to other long-context attention alternatives, which again can only be judged in practice.

---

> ### Author Rebuttal · Authors · 2025-07-31
>
> We thank the reviewer for the detailed comments. We address the questions and concerns below.
>
> **Empirical validation (and acknowledgement):** We agree that our paper gives rise to many interesting questions which should be the study of follow-up empirical work. However, there are two main reasons we chose not to include them here.
>
> First, we feel that the theory results presented in our paper stand on their own. The goal of our work here is to perform a complete theoretical analysis of how small transformers can be used to simulate big ones. It is important to nail down these precise dependencies before implementing the approach in practice, and that is exactly what we do here - we provide a complete and systematic analysis, including optimal and yet simple simulation algorithms. Most notably, some of our results show that surprisingly efficient simulations are possible (for example, Theorem 1.1 and Lemma 3.3 show that we can make use of all the layers of an oracle simultaneously), and we’re also able to rigorously pin down when techniques like attention sinks, sliding window, or assumptions on the data can yield real improvements.
>
> We also want to highlight that our results indicate that quadratic/linear amounts of small transformers have at least as much representational strength as a constant amount of large transformers in different settings. Such general representation results can only be proved theoretically.
>
> Second, an empirical validation of our results here that is truly meaningful would need to relate to real LLMs that are trained on large corpa of text. The implementation and parameter tuning can now build on our results, but this would be a substantial undertaking, and likely depends quite a bit on the setting (data, hardware etc like the reviewer suggests). Our simulation algorithms are simple and even “differentiable”, and therefore could be implemented effectively in practice.
>
> For these reasons, we feel that careful experimental work should comprise a second paper (or perhaps even more than one follow-up paper for different applications). We will modify the limitation section - thank you for pointing it out.
>
> **Whether modern GPUs can do faster transformer inference needs to be shown experimentally:** Our main argument here is that when processing an extremely long input sequence, it is possible to partition the input sequence into multiple segments and deal with them separately while still obtaining the correct inference output. For example, we prove that a single GPU that can only take input length of at most $M$ can still perform inference on sequences of $N\gg M$ by doing inference multiple times. It is not clear what a straightforward/trivial time (wall-clock, FLOPs, etc) to compare to even is, because it is initially unclear how inference can even be performed with such limited resources.
>
> In addition, the fact that transformer-specific GPUs allow faster inference on short-to-moderate input length has been experimentally verified and is a core idea for many startups, as we mentioned inline 31-32. For example, Etched’s Sohu ASIC claims to be 20x times faster than NVIDIA H100 on Llama 70B (we are not allowed to add links here due to Neurips rules, but the details can be found on Etched’s website).
>
> **Number of calls to different models could still be very large:** We are not sure if we fully understand what “different models” here mean. We only consider transformers, so as long as all the models are based on the transformer architecture, our analysis will hold. Our assumption here is that the small transformers have all the intricacies of the large ones: if the large transformer has rotary embedding, skip connection, layer normalization for example, the small ones can simulate it if they also have the same components.
>
> **Output approximation and bit precision:** Our main result (Theorem 1.1) provides an exact algorithm if we assume infinite (full) bit precision. The reason we define simulating as an approximation up to $O(1/2^{N})$ multiplicative error is that, as in most theoretical works in this area, we assume only $O(\log n)$ bit precision to more accurately model realistic hardware, which means that all parameters are represented by $O(\log N)$ bits. In this case, it is impossible to achieve perfect accuracy because the $exp$ function in the attention mechanism will have to be approximated. In other words: this approximation error is unavoidable in limited-precision hardware. Our results provide the best possible approximation in this scenario.
>
> **More insights on model size vs gains:** Our main results require around $2N^2/M^2$ small transformers (or calls) to simulate a large one, so in theory the larger $M$ is, the fewer inferences we need to perform. We agree that tuning $M$ (and other model parameters) and finding out the regime where we can obtain most gains is an interesting empirical problem to study in the future.
>
> **Is the core of the results Lemma 3.1 and comparison to Flash Attention:** This is a great question! Yes, Lemma 3.1 is one core of our proof (the other core being Lemma 3.3 where we deal with multiple heads and layers) and it shares some insights with Flash Attention, although they are still different.
>
> 1. Similarities: Both Flash Attention and our algorithm deal with a large attention matrix by decomposing it into smaller blocks.
>
> 2. Differences:  The goals are different: the major concern of Flash Attention is IO complexity, and here we study representational strength. In addition, the main technical challenge is different: in Flash Attention, you load query, key, matrix value blocks into SRAM and do whatever computation you can do on SRAM, while here we can only perform what small transformers can perform. Therefore, our model is more limited.
>
> **More insights on our algorithms:** We present proof sketches in section 3 for our main results. Although the algorithms for average-case inputs and attention sinks are slightly different, they share similar ideas. Proof sketch of Lemma 3.1 (starting from line 273) best summarizes our ideas. We will make the sketch clearer in the final version of this paper.
>
> We hope that our responses make sense, and make the reviewer believe that our work has its important value even though it is purely theoretical. Please let us know if there are any more questions!

---

> > ### Comment · Reviewer_BQJT · 2025-08-01
> >
> > Thank you for your detailed response.
> >
> > > We are not sure if we fully understand what “different models” here mean.
> >
> > By "different models," I was referring to models with distinct sets of weights; essentially highlighting the practical overhead involved in loading each model's weights onto the GPU for inference.
> >
> > > More insights on our algorithms: We present proof sketches in section 3 for our main results.
> >
> > Could you kindly clarify whether the arguments presented in the paper are purely existential proofs, or if there is a concrete algorithm that specifies how to simulate a reference model on input length N using multiple models on shorter inputs M? Specifically, does the algorithm provide the weights and inputs for each of the smaller models?
> >
> > > Empirical validation
> >
> > Thank you for the clarification. Based on your reasoning, I believe it would be more effective to motivate the work based on the representation equivalence angle, rather than the practical implications behind the work.

---

> > > ### Author Response · Authors · 2025-08-02
> > > **Response to Reviewer BQJT**
> > >
> > > Thank you for the clarification and questions!
> > >
> > > **Whether arguments are existential or construction**: Yes, our simulation algorithms are concrete algorithms: if we know all parameters of large transformers, our algorithms will find all the parameters in the small transformers such that they simulate the output of the large transformer. In fact, finding these parameters are not only computationally efficient, the query/key/value matrices in the small transformers are all very similar to the query/key/value matrices in the large one except that we pad one extra row/column (fixed in advance, independent of the parameters in the large transformer). The concrete construction can be found in supplementary material.
> > >
> > > **Empirical validation:** We still believe that our work is important both in terms of representation and practical efficiency because it lays the foundation for future (empirical) work. Making transformer inference faster is a difficult yet important task that requires significant efforts, and we provide a new, provably-efficient idea to achieve this goal. The results on representation are necessary for our purpose, but our results go beyond representation because our simulation algorithms are simple, construction-based and even differentiable. These characteristics show that simple implementation is possible.
> > > We agree that eventually we will need (a large amount of) empirical validation to achieve faster transformer inference for any idea, but we hope that the reviewer agrees with us that this paper provides a novel and provably-efficient idea that lays the foundation of future research along this line.

---

> > > > ### Comment · Reviewer_BQJT · 2025-08-05
> > > >
> > > > Thank you for your response. I think providing a clear and concrete description of the algorithm in the paper would be helpful for readers. Regarding the lack of experimental validation, while I acknowledge the authors' claim that they "provide a new, provably-efficient idea to [making transformer inference faster]" I, like other reviewers, still believe the paper should include at least some proof of concept to support its claims. Therefore, I will maintain my final rating.

---

### Official Review · Reviewer_tBwa · 2025-06-29

**Clarity:** 4
**Significance:** 2
**Originality:** 3
**Rating:** 4
**Confidence:** 3

**Summary:**

Evaluating large language models (LLMs) on lengthy sequences is computationally intensive due to the quadratic cost of their attention mechanisms. This work presents a method to address this challenge. The authors developed a theoretical framework that effectively simulates a large transformer with a sequence length of N by using O((N/M)^2) transformers, each processing a sequence of length M (where M is much smaller than N). Furthermore, they extended this framework to accommodate different attention mechanisms, such as sliding windows and attention sinks, which significantly reduces the requirement to just O(N/M) smaller transformers. They also found that, on average, a linear number of transformers are sufficient for the general case, provided that attention and value parameters are bounded.

**Questions:**

* The authors covered sliding windows and attention sinks, but they didn't mention the causal masking case. I assume it might follow the same quadratic scaling as the general case, but I'm curious if its specific structure could allow for a more efficient simulation.
* I'm curious about KV-caching; can it enhance the suggested algorithm?

**Ethical Concerns:**

["NO or VERY MINOR ethics concerns only"]

**Final Justification:**

Despite the extensive theoretical framework presented in the paper, I believe that additional empirical validation is necessary for the work to feel complete. As such, I will not be increasing my score.

**Limitations:**

yes

**Paper Formatting Concerns:**

no conerns

**Quality:**

3

**Strengths And Weaknesses:**

Strengths -
* This paper tackles a critical problem in the field of large language models. Creating efficient algorithms—perhaps using distributed computing—for LLM inference on long sequences could significantly boost performance.
* This paper is highly readable and well-structured. The theorems are presented clearly, and the proof sketches offer concise explanations of the authors' methodology.

Weaknesses -
* The paper would benefit significantly from empirical validation or a proof of concept. The oracle constructions proposed in Section 3 appear plausible, and some practical results would be highly valuable. For instance, an inference efficiency comparison across different values of M could effectively illustrate the trade-off between practical benefits and theoretical compexity. Additionally, it would be insightful to understand where real-world data falls on the spectrum between quadratic and linear scaling. Given the clear potential impact on practical applications, including some empirical results is crucial.

---

> ### Author Rebuttal · Authors · 2025-07-27
>
> We thank the reviewer for the insightful comments, and answer the questions below.
>
> **Empirical validation and parameter tuning:** We agree that our paper gives rise to many interesting questions which should be the study of follow-up empirical work. However, there are two main reasons we chose not to include them here.
>
> First, we feel that the theory results presented in our paper stand on their own. The goal of our work here is to perform a complete theoretical analysis of how small transformers can be used to simulate big ones. It is important to nail down these precise dependencies before implementing the approach in practice, and that is exactly what we do here - we provide a complete and systematic analysis, including optimal and yet simple simulation algorithms. Most notably, some of our results show that surprisingly efficient simulations are possible (for example, Theorem 1.1 and Lemma 3.3 show that we can make use of all the layers of an oracle simultaneously), and we’re also able to rigorously pin down when techniques like attention sinks, sliding window, or assumptions on the data can yield real improvements.
>
> We also want to highlight that our results indicate that quadratic/linear amounts of small transformers have at least as much representational strength as a constant amount of large transformers in different settings. Such general representation results can only be proved mathematically/theoretically.
>
> Second, an empirical validation of our results here that is truly meaningful would need to relate to real LLMs that are trained on large corpa of text. The implementation and parameter tuning can now build on our results, but this would be a substantial undertaking, and likely depends quite a bit on the setting (data, hardware etc). Our simulation algorithms are simple and even “differentiable”, and therefore could be implemented effectively in practice.
>
> For these reasons, we feel that careful experimental work should comprise a second paper (or perhaps even more than one follow-up paper for different applications).
>
> **Causal masking:** We state in our main Theorem (Theorem 1.1, line 107-108) that indeed, a quadratic amount of small transformers are able to simulate a large one, assuming they both have causal masking. A similar argument to the one in line 109-113 will show that this is optimal as well.
>
> **KV cache:** This is a great question! Our algorithms are indeed compatible with KV cache because it can be used in a standard way for small transformer inference. The trivial space needed will be $O(N^2/M^2)\cdot O(Md) = O(N^2d/M)$ for the main simulation. However, in our actual construction, we only need to store what needs to be stored in a large transformer inference (all keys and values) plus a constant amount of fixed vectors such as the all one vector. As a result, the space needed is $O(Nd)$, which is exactly the same as normal KV cache takes.
>
> We hope our responses make sense, and please let us know if there are any more questions!

---

> > ### Comment · Reviewer_tBwa · 2025-08-05
> >
> > Thank you for your detailed response and for clarifying my concerns regarding the causal masking and KV cache.
> >
> > I understand that the paper's primary focus is on the theoretical formulation of the simulation, and that adding extensive empirical validation would be beyond its scope. However, I still believe that a small proof-of-concept using a "smaller" LLM would have significantly strengthened the paper. Such a demonstration could validate the simulation and show the impact of different values of M and the effect of Theorem 4.1.
> >
> > You mentioned that the simulation is "differentiable." Does this mean it can be used for training?

---

> > > ### Author Response · Authors · 2025-08-05
> > >
> > > We thank the reviewer for the further engagement and feedback.
> > >
> > > Yes, it means that all the simulation processes are differentiable (such as padding numbers and feed the results to the small transformer) and therefore can be used for training. The whole procedure can be considered as a composition of padding and oracle inference, and they are both differentiable and trainable.

---

### Official Review · Reviewer_6Vvf · 2025-07-03

**Clarity:** 3
**Significance:** 3
**Originality:** 3
**Rating:** 5
**Confidence:** 2

**Summary:**

In this work, the authors propose to use small transformers to approximate the output of a large one. This study highlights:

1. This paper explores the representational power of small Transformers in simulating large Transformers that process long input sequences.

2. Such simulation can potentially save inference and training cost as modern GPUs are optimized for shorter sequence lengths.

3. A Transformer with input length $ N $ can be efficiently simulated using $ O\left((N/M)^2\right) $ smaller Transformers with input length $ M \ll N $, and this quadratic bound is tight in the worst case.

4. This conclusion could also be extended to transformers with sliding window attention, or attention sinks.

**Questions:**

1. In the study, the transformer model is simplified to merely containing MLP and Attn blocks. In a real transformer model, things are much more complex, such as rotary embedding, skip connection, layer normalization, etc. Should these partitions draw to a different conclusion?

**Ethical Concerns:**

["NO or VERY MINOR ethics concerns only"]

**Final Justification:**

I suggest Accept. My primary considerations about this paper are:

1. This paper is well written. It focuses on a very important topic (smal vs. large model) which still lacks theoretical analysis. I believe that the meaning behind this paper is the most important.
2. Questions on why the authors apply such simplicity and why the simplicity is reasonable are clearly addressed by the authors. This is important because this indicates that the assumption closely related to reality.
3. Although empirical validation is not included (as pointed out by some reviewers), I believe this may go beyond the domain of this paper. Lacking validation would not diminish the contribution of this work. I think this is not a main drawback.

**Limitations:**

yes

**Paper Formatting Concerns:**

1. The abstract must be limited to one paragraph.

**Quality:**

3

**Strengths And Weaknesses:**

Strengths:
1. This paper is well written and easy to understand. The notations and equations are self-consistent and conclusions are presented in clear format.
2. It is innovative to simulate a large transformer with smaller ones. The authors made remarkable progress compared to related works.

---

> ### Author Rebuttal · Authors · 2025-07-26
>
> We thank the reviewer for the thoughtful comments, and answer the questions below.
>
>
> **Transformer model simplification:** This is a great question! We simplify our transformer model for the sake of simpler theoretical analysis, but in general these additional components will not affect our conclusion, and the reason is that whatever component we add to the large transformer, we can also add it to the small transformers in the same way to ensure that our analysis still holds.
> 1. Rotary embedding: When a large transformer has rotary embedding, each query and key is “rotated” by some fixed rotation function in each attention head. The same rotation function can be applied to each query and key in attention heads in small transformers as well.
> 2. Skip connection: One can usually think of skip connection as if we have an additional attention head (let’s call it skip attention head) in each layer whose output is identical to input. This additional attention head can be easily simulated by $N/M$ smaller skip attention head because we can feed the input of length $N$ to $N/M$ many smaller skip attention heads. Therefore, if both large and small transformers have skip connections, a large one will still be simulated by small ones (number needed is still the same).
> 3. Layer normalization: Layer normalization will not affect our analysis because whatever normalization that is performed in large transformers, we can do the exact same in small transformers. Therefore, if both large and small transformers have layer normalization, a large one will still be simulated by small ones (number needed is still the same).
> We will make sure we mention this in the final version of the paper.
>
> **Abstract is limited to one paragraph:** Thank you for pointing this out, and we will make sure we fix it in our final version.
>
> We hope that our answers make sense, and make the reviewer more confident about the value of our work. Please let us know if there are any more questions!

---

> > ### Comment · Reviewer_6Vvf · 2025-08-04
> > **Thank you**
> >
> > Thank you for your response. The simplification of transformer model and the corresponding analysis make sense. After considering the comments from other reviewers and your responses, I have decided to maintain my score (borderline accept).

---

> ### Comment · Reviewer_6Vvf · 2025-08-09
> **Update on final justification**
>
> Upon careful thoughts on the authors' rebuttal and discussions in the fellow reviewers, I decide to raise my score to Accept and my  confidence to 2. Here are my primary considerations:
>
> 1. Small models are significant in LLM research and application. The alignment of small and large models is popular, which gives rise to many frontier topics, such as model distillation, speculative decoding, etc. This paper reveals a very interesting aspect of alignment, and the authors successfully provide theoretical guarantee for this. This approach may offer new solutions to many cutting-edge problems, not only accelerating inference.
> 2. The author successfully identify the primary and secondary components within the complex transformer architecture and simplify it accordingly. This simplification is effective for other components as well, such as rotary embedding, skip connections, layer normalization, and KV-cache. This indicates that the assumption about the problem is not detached from reality.
> 3. The method proposed by the authors can save computation. However, during the model inference, things are relatively complex, and the inference process is influenced by various factors. I still encourage the authors to provide some numerical results and wall clock speedup, although these may go beyond the domain of this paper and is not considered a huge drawback for the theoretical analysis.

---

### Note · Authors · 2025-08-11

We thank all the reviewers for their insightful comments. We would like to briefly summarize our discussions and highlight the important aspects discussed.

**Theoretical side:** We believe that we have addressed all the concerns on the theoretical side that reviewers pointed out:
1. Our results still hold with extra components such as rotary embedding, layer normalization and skip connection etc.
2. Our algorithms are compatible with KV cache and basically have the same space usage.
3. Our algorithms are exact if we assume infinite bit precision, and are the best possible if we assume finite bit precision (say $O(\log N)$).
4. Our algorithms are “differentiable” and thus can be trained.

As reviewer 6Vvf suggests, our results are not detached from reality for these reasons. We are delighted to see that all the reviewers find our theoretical results interesting.

**Practical side:** Reviewers are concerned about the fact that this paper has a practical motivation but is purely theoretical.
1. From a conference perspective, we would like to mention that NeurIPS’s call for papers welcomes (purely) theory papers and there have been a lot of interesting theory papers without any experiment in NeurIPS. We believe that such theoretical results are important and interesting on their own (just as we believe that many purely empirical results are interesting on their own even though they do not have a proof of learnability).
2. We agree with reviewers that experiments will be exciting future work. As we mentioned in our rebuttal, we believe that such experiments would be substantial work that could result in a few papers. Therefore, we provide a provably-efficient framework for all the experiments that we hope will follow from. We believe that a complete, accessible theoretical framework is important for all followup work.
3. Our results on representation can only be shown theoretically.
4. We appreciate reviewer 6Vvf’s comments on further implications of our work. Indeed, we believe that our work has interesting connections with alignment. For example, our average-case results imply that it is possible for smaller models to align well with larger ones with certain assumptions on input data.

---

### Decision · Program_Chairs · 2025-09-17

**Decision:**

Accept (spotlight)

**Comment:**

This work considered the question of whether transformers with long sequences can be efficiently simulated by transformers that can only take short input sequences. It proved that any transformer with input length N can be efficiently simulated by O((N/M)^2) transformers with input length M, and proved that this is the best bound in the worst case. It then considered several beyond-worst-case scenarios and showed the optimal bound O(N/M).

The work proposed a novel and interesting question (simulating transformers on long inputs via transformers on short inputs), which can also be relevant to practice since it points to a potential new direction for acceleration (though not concrete practical algorithmic ideas).
The theoretical analysis is thorough, giving optimal bounds in the worst case, and also optimal bounds in several beyond-worst-case scenarios that may be relevant to practice.

Overall, the work is a solid theoretical work: asking an interesting question along an important direction and giving a thorough analysis with optimal bounds. The fact that the theoretical question/results are relevant to practice should be regarded as an additional benefit rather than a drawback.